# Phage-inducible chromosomal minimalist islands (PICMIs), a novel family of small marine satellites of virulent phages

Rubén Barcia-Cruz[1,2], David Goudenège[1,3], Jorge A. Moura de Sousa[4], Damien Piel[1,3], Martial Marbouty [5], Eduardo P. C. Rocha[4] & Frédérique Le Roux [1,3,6] ✉

Phage satellites are bacterial genetic elements that co-opt phage machinery for their own dissemination. Here we identify a family of satellites, named Phage-Inducible Chromosomal Minimalist Islands (PICMIs), that are broadly distributed in marine bacteria of the family Vibrionaceae. A typical PICMI is characterized by reduced gene content, does not encode genes for capsid remodelling, and packages its DNA as a concatemer. PICMIs integrate in the bacterial host genome next to the fis regulator, and encode three core proteins necessary for excision and replication. PICMIs are dependent on virulent phage particles to spread to other bacteria, and protect their hosts from other competitive phages without interfering with their helper phage. Thus, our work broadens our understanding of phage satellites and narrows down the minimal number of functions necessary to hijack a tailed phage.

Bacteriophages (or phages) are viruses that infect bacteria and may be the most diverse and abundant biological entities in the ocean[1,2]. Their ability to kill the host makes them key modulators of bacterial abundance and genetic diversity. Phages are themselves exploited by phage satellites, a class of mobile genetic elements that hijack the phage machinery to promote their own dissemination while interfering with phage reproduction[3–6]. Recently, several studies revealed that marine bacteria transduce a vast diversity of chromosomal islands, many of which might be satellites. These elements were named Virion Encapsidated Integrative Mobile Element (VEIME)[7] and Tycheposons[8]. Although these studies suggest that satellites play important roles in natural environments, the function and characteristics of these mobile elements are still poorly understood.

Satellites may develop different strategies for hijacking the life cycle of a helper phage, and the characterization of new satellites can reveal features that unite or separate different satellite families[9]. Our understanding of phage satellite lifestyles largely stems from a limited number found in clinically relevant bacteria. Phage-inducible chromosomal islands (PICIs)[3,10] are the most widespread satellites, being found in many Firmicutes and Gammaproteobacteria. They are closely related to the capsid-forming PICIs (cf-PICIs)[11,12], which are less abundant and primarily found in Enterobacteriaceae, Lactobacilli, Enterococci, Pasteurella, Bacillaceae, Streptococci, and Morganellaceae. P4-like satellites are prevalent in Enterobacteriaceae, Yersiniaceae, Erwiniaceae, Hafniaceae, and Pectobacteria[13]. In contrast, phage-inducible chromosomal islands-like elements (PLEs) are identified exclusively in *Vibrio cholerae*[14,15]. A feature common to all known satellites is the integration in specific sites of the bacterial chromosome. Excision is promoted by the induction of a helper prophage or infection by a helper phage and requires integrases and excision factors encoded by the satellite[6]. The circularized extrachromosomal element then replicates using its own origin of replication and is

[1]Sorbonne Université, CNRS, UMR 8227, Integrative Biology of Marine Models, Station Biologique de Roscoff, CS 90074, F-29688 Roscoff cedex, France. [2]Department of Microbiology and Parasitology, CIBUS-Faculty of Biology, Universidade de Santiago de Compostela, Santiago de Compostela, Spain. [3]Ifremer, Unité Physiologie Fonctionnelle des Organismes Marins, ZI de la Pointe du Diable, CS 10070, F-29280 Plouzané, France. [4]Institut Pasteur, Université Paris Cité, CNRS UMR3525, Microbial Evolutionary Genomics, Paris, France. [5]Institut Pasteur, Université Paris Cité, Organization and Dynamics of Viral Genomes Group, CNRS UMR 3525, Paris F-75015, France. [6]Département de microbiologie, infectiologie et immunologie, Université de Montréal, Montréal, Canada. ✉e-mail: frederique.le.roux@umontreal.ca

packaged into viral particles using *pac* and/or *cos* packaging systems[16]. Known satellites typically have genomes about one-third of the size of the helper phages. Many satellites encode mechanisms of capsid remodeling to fit their small genomes, whilst excluding the larger genomes of their helper phages[6,13,15,17]. Specific features of each phage satellite family include the lifestyle of their typical helper phage, i.e., temperate for the helpers of PICI, cf-PICI, and P4-like satellites vs. virulent for the helper phage of PLE named ICP1. Their genome sizes also vary, with an average of 10 (P4), 9.5 (PICI), 14 (cf-PICI), or 18 kbp (PLE). These differences in size are directly proportional to their gene repertoires and are associated with different mechanisms for the subversion of the host phage particles[12]. Their small size raises the question of the minimal gene set required for satellite excision, replication, and phage hijacking.

To understand the evolution of the interaction between the bacterial host, the phage, and the satellite, it is essential to know if the latter has a negative impact on the fitness of the others. The effect of satellites on phage reproduction varies across families, and sometimes even across satellites of the same family. While P4-like satellites and PICIs only interfere partially with the reproduction of their helper phages[18,19], PLEs completely abrogate the production of ICP1 progeny[15]. Finally, some P4-like satellites[20] and PICIs[21] encode hotspots of antiviral systems protecting both the bacterial host and their helper phages from competing phages and other mobile genetic elements. The associations of known phage satellites thus range from pure parasitism to mutualism in relation to their bacterial and phage hosts. While it has become clear that phage satellites are abundant and play diverse roles that affect their hosts, only a limited number have been described, hindering our ability to fully understand the breadth of their influence.

Given the abundance, diversity, and distribution of viruses in the ocean, identifying new phage satellites and understanding their functions might be akin to "looking for a needle in a haystack". Characterizing and establishing the functions of new phage satellites requires identification of the cognate helper phages and of the cellular host. In the present study, we addressed these challenges by taking advantage of bacteria from the *Vibrionaceae* family, which is notable for its extensive culture and sequence coverage of hosts and phages[22,23]. The *Vibrionaceae* (also referred to as 'vibrios') comprise a diverse group of bacteria that are widespread within marine environments, encompassing human and animal pathogens[24,25]. Vibrios are easily cultured, allowing isolation of their infective phages, whole genome sequencing, and inverse genetics.

We report the discovery of a new family of satellites that hijack virulent phages. We named this family of satellites PICMI (Phage-Inducible Chromosomal Minimalist Island) because of their reduced size and gene content. When sequencing the genome of a virulent phage and its *Vibrio chagasii* host, we detected concatemeric repeat sequences of PICMI in virus particles. PICMI encodes three core proteins necessary for excision and replication but is completely dependent on its helper phage to generate the PICMI infective particles and mobilize across bacteria. Despite this high dependency, PICMI does not strongly interfere with the fitness of its helper phage. However, the satellite can confer host protection from other phages. PICMIs are broadly distributed in the *Vibrionaceae* which encompasses the potential pathogens *V. cholerae*, *V. parahaemolyticus* and *V. vulnificus*[24], suggesting an important role of these marine phage satellites in vibrios.

## Results
### Identification of a satellite, its host, and the helper phage
We previously isolated and sequenced 49 phages infecting strains of *V. chagasii*[22]. One virulent phage (115_E_34.1, named Φ115 for simplicity) piqued our curiosity because its genome sequence assembly revealed two contigs of 47,851 and 6110 bp (Supplementary Fig. 1). The number of sequencing reads was 634,740 and 108,219 for the large and small contig respectively, with only 16 hybrid reads, providing evidence for two distinct mobile genetic elements in the Φ115 phage progeny. The large element corresponds to the genome of the Φ115 phage. The small element contains six genes encoding an integrase (*int*), a putative regulator (*alpA*), a putative primase (*prim*), and three other genes of unknown function (Fig. 1a). Among the genes of unknown function, one gene has four nucleotides overlap (ATGA) with the sequence of *int* and was named *iolg* for integrase overlapping gene. The two other genes were named *up1* and *up2* for unknown protein 1 and 2. Using Phanotate[26], a tool dedicated to phage genome annotation (see Methods), we identified six additional open reading frames (ORFs) (Supplementary Fig. 2), which were considered questionable because they encoded for proteins of 32–44 amino acids. The 6,110 bp element was also found in the genome of the host used to isolate the phage, *V. chagasii* 34_P_115 (herein named V115). Surrounded by two 17 bp direct repeats, this element integrates downstream of the *fis* gene (Fig. 1a). The *fis* gene encodes a DNA-binding protein that plays a crucial role in the efficient excision of phage lambda from the *Escherichia coli* chromosome[27]. Given these results, we posited that the 6,110 bp element is a satellite of the helper phage Φ115.

Despite the small size of the satellite (~1/8 the phage genome), we did not observe smaller-sized capsids commensurate to its genome size, as described for other known families of phage satellites (Supplementary Fig. 3). We tested for the presence of physical contacts between the two genomes[28] to confirm that the satellite was located in viral particles lacking the phage genome. We applied HiC[29] on different mixes of phage particles (see Methods). The result showed a clear absence of physical contact between the DNAs of the phage Φ115 and the satellite (Supplementary Fig. 4), confirming the exclusive packaging of the satellite in full-sized phage-like particles. To understand if the satellite fills the capsid by packaging as a concatemer, we performed single-molecule nanopore sequencing of Φ115- encapsidated high-molecular weight DNA and found a fraction of the viral particles contained a concatemer of 8 copies of the satellite of ~49 kbp size, similar to the Φ115 genome (Supplementary Fig. 5, Supplementary Data 1). Finally, we confirmed by Southern blot that concatemeric repeat sequences of the satellite show DNA of similar size to that of the genome of phage Φ115 packaged in viral particles (Fig. 1b). To estimate the percentage of Φ115 particles that contain the satellites instead of phage DNA, we first normalized the Illumina sequencing reads on genome size ($634{,}740/47{,}851 = 13.26$ for the phage and $108{,}219/6110 = 17.71$ for the satellite) and next considered that 8 copies of the satellite are packaged as concatemer in particles ($17.71/8 = 2.21$). This led to 16% ($2.21/13.26 * 100$) of the population being hitchhiked by the satellite. This estimation was further confirmed by Nanopore sequencing (10 or 15% depending on the replicate, Supplementary Data 1) and quantitative PCR (qPCR) of phage DNA (15%, Supplementary Fig. 6). Altogether, these data strongly suggest the identification of a new marine phage satellite. Due to its reduced size, we named this satellite PICMI$_{115}$, for Phage-Inducible Chromosomal Minimalist Island identified in *V. chagasii* strain V115 and its cognate helper phage Φ115.

### PICMI sustains minimal function of excision and replication
Our cultivation-enabled model system allowed to dissect the various steps in the PICMI's life cycle: (i) excision, (ii) replication, (iii) packaging, and (iv) transduction to a new host. Of these, we were not able to identify *cos* or *pac* packaging sites in the helper phage genome, or any homologs of genes involved in redirecting packaging that are characteristic of other satellite families (i.e., *terS*, *sid*, *ppi*). However, Nanopore sequencing of the viral particles revealed random extremities of the PICMI$_{115}$ concatemer (Supplementary Fig. 5), which is suggestive of a mechanism of headful (*pac*-like) packaging.

Before investigating the induction of PICMI$_{115}$ by the helper phage, we produced Φ115 viral particles without the satellite (designated as Φ115pure) by generating phages from a V115 derivative that

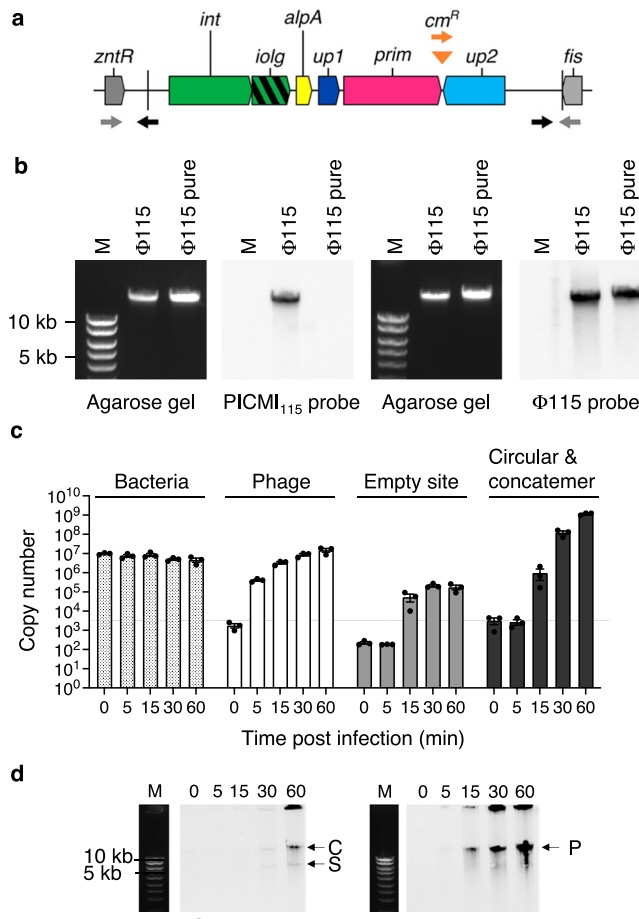

**Fig. 1 | Excision and replication dynamics of PICMI₁₁₅. a** Schematic representation of PICMI₁₁₅ integrated between the *zntR* and *fis* genes encoding a HTH-type transcriptional regulator and the factor for inversion stimulation respectively. Arrows depict inward- and outward-directed primers used to detect the integration site after excision (gray) or the single circular and concatemeric forms of PICMI₁₁₅ (black). For transduction experiments PICMI₁₁₅ was marked by Cm$^R$ (orange triangle). Orange forward and gray reverse primers were used to control of the integration of PICMI₁₁₅ at the end of *fis* gene. **b** Estimation of phage and PICMI DNA size. DNA extracted from phage Φ115 or as control Φ115pure, were separated on a SYBR Green stained agarose gel and Southern blotted with PICMI₁₁₅ or Φ115 probes. M: molecular marker (Smart ladder Eugentec). Images are representative of two independent experiments. Uncropped gels and blots are provided as a Source Data file. The dynamics of excision and replication following phage Φ115 infection were explored by qPCR (**c**) and Southern blot (**d**). **c** Standard curves allowed for the determination of the number of DNA copies per 20 ng of DNA. Source data are provided as a Source Data file. Bar charts show the mean ± Standard Error of Mean (SEM) with $n = 3$ independent replicates depicted by individual dots. The dashed line signifies the limit of detection for this assay. Spearman's test for *x* versus time (*p*-value) indicates a significant increase in copy at 15 min for phage (0.9602, $p < 0.001$), the empty site (0.8183, $p = 0.0002$), and circular/concatemer PICMI (0.9274, $p < 0.001$). In contrast, a reversed trend is observed for vibrio (0.7419, $p = 0.0015$). At the 30-minute mark, a one-tailed *t*-test (*p*-value), transformed to Log10, demonstrated significance for phage (0.0060) and circular/concatemer PICMI (0.0059) only. **d** In the Southern blot C, S, and P indicate the concatemeric and single form of PICMI₁₁₅ and phage genome respectively. Images are representative of two independent experiments.

lacked the entire PICMI₁₁₅ (ΔPICMI₁₁₅). The absence of the satellite in the Φ115pure population was confirmed by Nanopore sequencing (Supplementary Data 1), qPCR (Supplementary Fig. 6), and Southern blot (Fig. 1b). We then proceeded to investigate the kinetics of PICMI₁₁₅ activation following V115 infection by Φ115pure. Inward-directed primers (as depicted in Fig. 1a) were utilized in qPCR to detect the empty

integration site resulting from PICMI₁₁₅ excision. Outward-directed primers were employed to quantify the circularized PICMI₁₁₅, either as a single circular form or in concatemeric form. Amplicons were obtained 15 min after adding the phage to the bacterial culture (Fig. 1c), the estimated time for its complete adsorption and phage DNA injection in the host cytoplasm (Supplementary Fig. 7). PICMI₁₁₅ activation was observed exclusively in the presence of the helper phage (Supplementary Fig. 8). A dramatic increase of the copy number of the circular and/or concatemer DNA was observed 30 min after phage addition (Fig. 1c), indicating intensive replication. The increase in the amount of a DNA band at the size of the concatemer is observed by Southern blot at 60 min (Fig. 1d), consistent with a plasmid-like rolling circle DNA replication mechanism.

It is expected that PICMI₁₁₅ transduction requires the viral particles to adsorb on the recipient host. The phage Φ115 was previously described as having a narrow host range[22], with only 2 out of 136 *V. chagasii* strains (V115 and V157) susceptible to Φ115 infection and reproduction. These two strains each encode one (identical) PICMI. Testing closer phylogenetic neighbors to the original host V115, we found that the phage Φ115 adsorbs to the strain V511 without producing progeny (Supplementary Fig. 9), probably due to intracellular defense mechanisms[23,30]. Specifically, the Dnd and Lamassu defense systems were identified in V511 using Defense-Finder[31], while they were notably absent from V115. The strain V511 does not carry PICMI and shows 100% identity with the *fis* gene of V115 and V157. We thus assumed that the strain V511, in addition to the V115 derivative lacking PICMI₁₁₅ (ΔPICMI₁₁₅), could be used as recipient for transduction assays. We first inserted a chloramphenicol resistance marker (Cm$^R$) downstream of the *up2* gene of PICMI₁₁₅ (Fig. 1a) and infected this strain with Φ115pure to produce lysates of viral particles with the Cm$^R$-encoding PICMI₁₁₅. The introduction of the Cm$^R$ cassette increased the percentage of PICMI₁₁₅ in phage particles (Supplementary Fig. 6). We thus assumed that the excision, replication, and packaging functions of the PICMI₁₁₅-Cm$^R$ satellite were intact. We next used serial dilutions of this lysate to infect the two susceptible hosts (ΔPICMI₁₁₅ and V511) and selected for chloramphenicol resistant cells that acquired the PICMI₁₁₅-Cm$^R$ satellite. Transductant Forming Units, TFU, were obtained using a lower titer of phages for ΔPICMI ($10^5$ Plaque Forming Unit, PFU, ml$^{-1}$; Multiplicity of infection, MOI 0.01) compared to V511 ($10^8$ PFU ml$^{-1}$; MOI 10). Consequently, the ratio between TFU and PFU was much higher with ΔPICMI (~$6.10^{-2}$) than with V511 (~$10^{-5}$; Fig. 2a). We confirmed by PCR that the integration of PICMI₁₁₅-Cm$^R$ occurred downstream of the *fis* gene in all tested transductants (Fig. 2b). Altogether our results showed that PICMI₁₁₅ is activated by a virulent phage, replicated by rolling circle, packaged, and transduced as a concatemer. With roughly 15% of viral particles that contain the satellite, the question arises whether PICMI₁₁₅ inflicts a cost on its helper phage.

Most known satellites interfere at least partially with the reproduction of their helper phage, although this effect can vary broadly between families[15,18,19]. We thus quantified the extent to which PICMI₁₁₅ interferes with its helper phage Φ115. We compared the titer of phages produced by the bacterial strains V115 wild type (wt), ΔPICMI₁₁₅, and two clones of ΔPICMI₁₁₅ + PICMI₁₁₅-Cm$^R$ after infection by Φ115pure. No significant differences (ANOVA, $p = 0.74$, *F*-test) were observed between the strains (Fig. 2c), showing that PICMI₁₁₅ does not strongly impact the reproductive fitness of its helper phage.

## AlpA is a key regulator of PICMI activation

Having established that PICMI₁₁₅ is a phage satellite, we further analyzed the role of each PICMI₁₁₅-encoded gene in the various aspects of the satellite's lifestyle. Each of the six genes (Fig. 1a) was deleted in V115, and the mutants were compared to the wild-type host for excision, packaging, and transduction. This revealed that *alpA, int*, and *iolg* are necessary for PICMI₁₁₅ excision induced by the helper phage

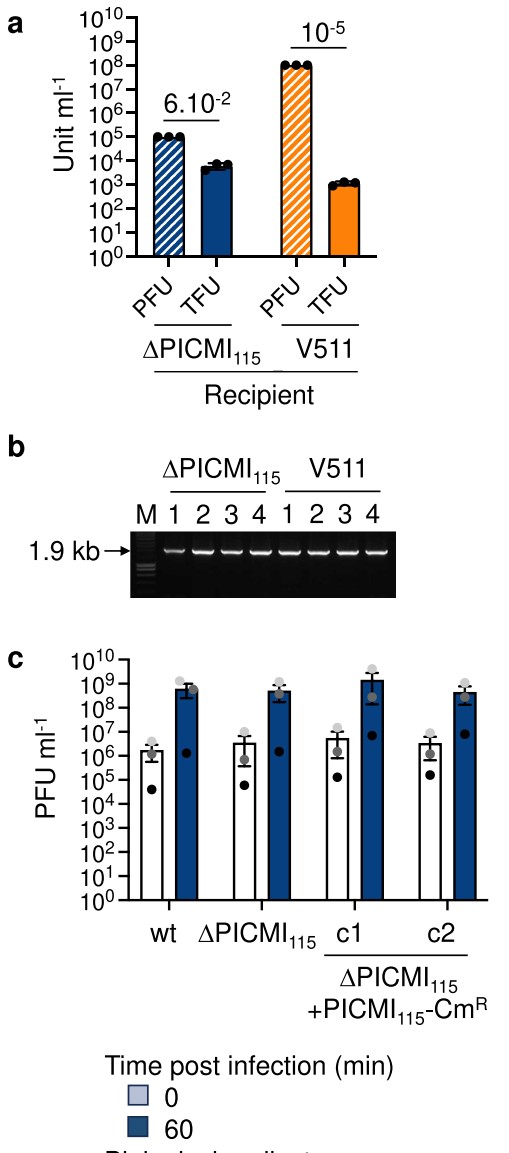

**Fig. 2 | Analysis of transduction and interference induced by PICMI$_{115}$. a** To determine transduction frequencies, lysates containing viral particles with the chloramphenicol-resistant (Cm$^R$)-encoding PICMI$_{115}$ were used at the specified titer (PFU ml$^{-1}$) to infect both a derivative lacking the full PICMI$_{115}$ ($\Delta$PICMI$_{115}$) and V511 hosts. Transductants (TFU ml$^{-1}$) were selected on chloramphenicol. Bar charts show the mean values ± standard deviation (SD) with $n = 3$ independent replicates depicted by individual dots. The average TFU/PFU ratio is indicated for each strain. **b** The integration of PICMI$_{115}$-Cm$^R$ at the end of the *fis* gene was confirmed by PCR. Images are representative of two independent experiments. Uncropped gel is provided as a Source Data file. **c** To investigate interference with the reproduction of the helper phage, the wild-type V511 strain, $\Delta$PICMI$_{115}$, and two clones (c1 and c2) of transductants carrying PICMI$_{115}$-Cm$^R$, were infected with $\Phi$115pure. The bar charts depict the phage titer at 0 and 60 min post-infection, with values shown the mean ± SEM from three independent experiments (depicted by individual dots). At 60 min, the null hypothesis (H0) stating that the values are similar across categories cannot be rejected. The results of the t2 Wilcoxon test yielded a *p*-value of 0.9957. Additionally, ANOVA analysis with an F-value of 4.227 and a *p*-value of 0.7419 further supports the acceptance of H0. The Tukey-HSD test for every pair did not reveal any significant differences ($p > 0.05$).

(Fig. 3a and Supplementary Fig. 10). The deletion of *prim* results in a lower number of circularized/concatemeric PICMI$_{115}$ copies, relative to the wild type, indicating that the primase is involved in PICMI$_{115}$ replication (Fig. 3a and Supplementary Fig. 10). Accordingly, the number of viral particles that contain PICMI$_{115}$ in phage progenies was strongly reduced in $\Delta alpA$, $\Delta int$, $\Delta iolg$, and $\Delta prim$ deletion mutants (Fig. 3b, Supplementary Fig. 6). It also led to a much lower transduction of PICMI$_{115}$-Cm$^R$, below our limit of detection (Table 1). When expressed in *trans* in the V115 derivative $\Delta int$, *int*, with (Fig. 3c) or without (Supplementary Fig. 11), the overlapping gene *iolg*, restored the helper phage-induced excision of PICMI$_{115}$. The expression of *int/iolg* also complemented the $\Delta iolg$ deletion (Fig. 3c). Notably, the expression of *alpA* in trans was sufficient to induce PICMI$_{115}$ excision even in the absence of phage and the copy number of circularized satellites increased 60 min post infection (Fig. 3c).

AlpA is predicted to act as a DNA-binding regulator, and previous work suggested that it is a transcriptional regulator[3] and/or an excisionase[32]. In PICMI$_{115}$, the expression of *alpA* from a plasmid did not alter the expression of the satellites' genes, suggesting that *alpA* induction of PICMI$_{115}$ excision is not mediated by transcriptional regulation (Supplementary Fig. 12a, b). We used ColabFold[33] to search for structural similarities with known excisionases and found strong similarities with the TorI regulator of *E. coli* and the Xis excisionase of *Streptomyces ambofaciens* (Supplementary Fig. 12c). We propose that among the three genes essential for excision, *int* and *iolg* are constitutively expressed, *alpA* is activated by phage, and the three proteins are involved in the formation of the excision complex.

**The intensity of PICMI$_{115}$ activation depends on the helper phage**
The induction of the satellite can be highly specific to the helper phage(s)[5]. We searched for other phages that could infect the strain V115 to establish whether induction of PICMI$_{115}$ is also specific to the helper phages $\Phi$115. We used viruses from seawater collected at the same oyster farm four years later (see Methods) and isolated seven phages infecting the host V115. The probe directed against phage $\Phi$115 also detected the newly isolated phages by Southern blot, suggesting that they are genetically related (Fig. 4a). Except for $\Phi$27, PICMI$_{115}$ was detected in the progenies resulting from the infection of all other phages, although at a much lower quantity than in the $\Phi$115pure infection (Fig. 4a). The phage $\Phi$27 was unable to induce a detectable excision of PICMI$_{115}$ (Fig. 4b). We thus compared the expression of the PICMI genes upon $\Phi$115pure (Fig. 4c) and $\Phi$27 infection (Fig. 4d). This revealed that 15 min post infection by $\Phi$115pure, presumably as soon as the phage is injected in the cytoplasm (Supplementary Fig. 7), only *alpA*, *up1* and *prim* were upregulated. Hence, these three genes define an early regulon. An increase of transcripts for the remaining PICMI$_{115}$ genes (*int, iolg,* and *up2*) was observed after 60 min. The latter might result from the increase in copy number of the PICMI$_{115}$ genome and not necessarily through the activation of gene expression. Infection by phage $\Phi$27, which did not lead to PICMI$_{115}$ induction, has no reproducible effect on PICMI$_{115}$ gene expression (Fig. 4d). Altogether, our results demonstrate that $\Phi$115 spurs strong induction of PICMI$_{115}$ and its ability to induce PICMI is related to the induction of the early PICMI$_{115}$ regulon.

**PICMI-like elements are broadly distributed in the Vibrionaceae**
Our description of the PICMI minimalism was based on a single model system identified in our collection, raising questions about the size, distribution, and diversity of PICMI-like elements in bacterial genomes. We thus built a MacSyFinder model[34] to allow the automatic identification of this element in bacterial genomes. Our PICMI$_{115}$ prototype is integrated in the V115 host genome downstream of the *fis* regulator

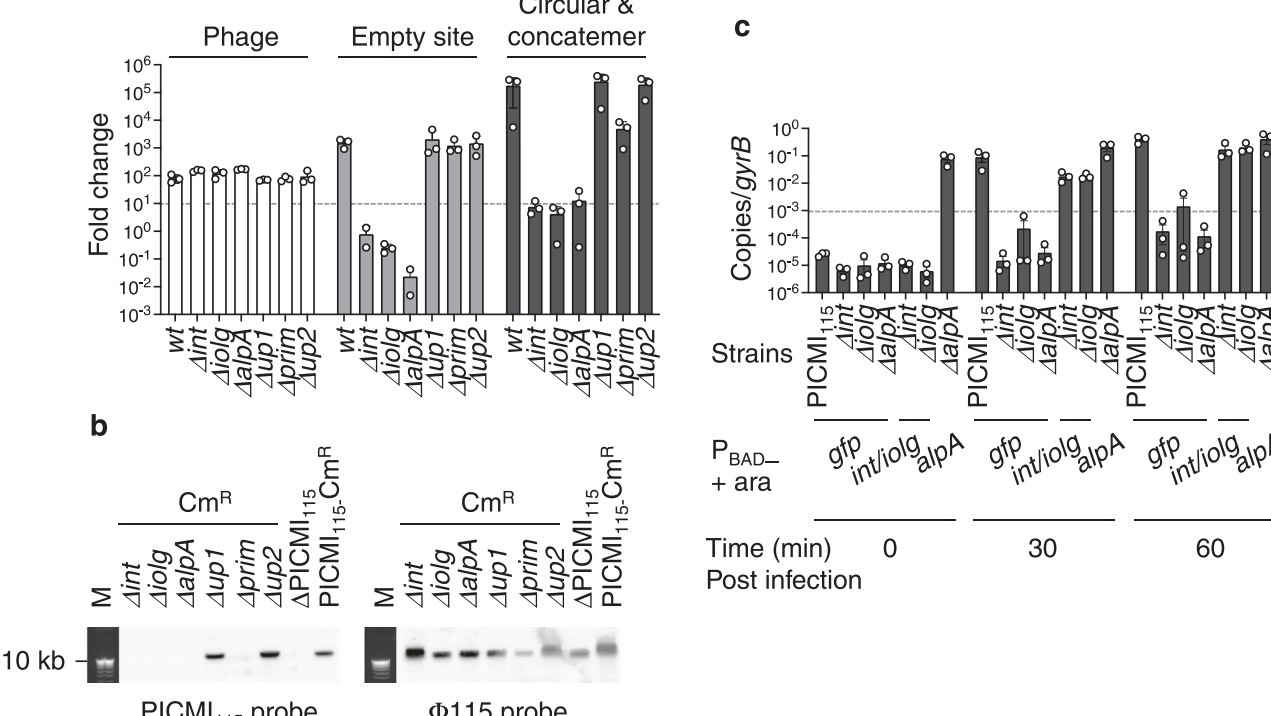

**Fig. 3 | Genes involved in PICMI₁₁₅ activation. a** Quantification of PICMI₁₁₅ activation after infection by Φ115pure in wild-type and single-gene mutants (e.g. *Δint*) was performed using qPCR. Standard curves facilitated the determination of DNA copies per 20 ng of DNA, normalized by bacterial copies (*gyrB*) per sample. Source data are provided as a Source Data file. Fold change was calculated by comparing DNA samples collected 30 min post-infection to the initial input before phage addition. The dashed line indicates the limit of detection for this assay. Bar charts represent the mean ± SEM from three independent experiments (individual dots). In the case of circular/concatemer PICMI and empty site, ANOVA revealed a *p*-value of <0.0001, and Tukey's test indicated that *Δint, Δiolg ΔalpA*, were significantly different from the wild type (*p* < 0.05). For Phage, ANOVA yielded a *p*-value of <0.0069, and the Tukey test showed that none were different from the wild type (*p* > 0.05). **b** Phage progenies were produced using a derivative lacking the full PICMI₁₁₅ (ΔPICMI₁₁₅), a marked PICMI (PICMI₁₁₅-CmR) and derivatives lacking a single gene as host. Phage DNAs were separated on an agarose gel and Southern blotted with PICMI₁₁₅ or Φ115 probes. M: molecular marker (Smart ladder Eugentec). Images are representative of two independent experiments. Uncropped gel and blots are provided as a Source Data file. **c** For complementation assays, the PICMI₁₁₅ genes (*int/iolg* and *alpA*), or as a control, *gfp*, were cloned under the conditional P_BAD promoter. The resulting plasmids were transferred into the respective mutants. Strains were cultivated in the presence of 0.2% arabinose for P_BAD induction and subsequently infected with Φ115pure for the indicated duration. The number of circular/concatemeric copies of PICMI₁₁₅ was quantified by qPCR (Source Data file) and normalized to the *Vibrio gyrB* copy number per sample. Bar charts depict the mean ± SEM from three independent experiments (individual dots). Notably, the expression of *alpA* induces PICMI₁₁₅ activation in the absence of phage (strain *ΔalpA* + P_BAD-*alpA* in the presence of arabinose at time 0). A one-tailed t-student's test comparing the *ΔalpA* complemented strain between 0 min and 30 min was not significant, while it showed significance (*p* < 0.05) at 60 min.

gene and contains only six genes. Among those, we showed that *int*, *alpA*, and *prim* are necessary for PICMI₁₁₅ lifestyle. In accordance with the experimental data, we set the presence and colocalization of *int*, *alpA* and *prim* genes as mandatory in the model, whilst adding *fis* as an optional gene marker in order to search for PICMI that use other integration sites. We then used it to search for PICMI-like elements in all Genbank bacterial complete genomes (v243, 05/26/2021) and identified 135 elements (Supplementary Data 2). From this list, the 67 satellites in *Vibrionaceae* genomes have a significantly smaller size than the others (average 6.7 kbp, unpaired t test *p* < 0.0001) (Supplementary Fig. 13).

We extended our search for putative PICMIs to a much larger dataset including all the available 19185 *Vibrionaceae* (drafts and complete genomes NCBI Assembly database, 02/16/2023). Of note, although we find incomplete PICMI-like variants across different bacterial species, we have not found complete PICMI other than in *Vibrio* genomes. We identified a total of 97 elements in diverse *Vibrionaceae* species (Fig. 5, Supplementary Data 3). We never detected more than one PICMI-like element in a genome, contrasting with P4-like satellites in *E. coli* genomes, that can contain up to three of the latter elements[13].

We also never detected any PICMI that did not include also *fis* (Supplementary Fig. 14). Hence, integration at *fis* is a hallmark characteristic of PICMI. Pairwise alignment of the DNA sequences of PICMI-like satellites permitted grouping them into 35 distinct subfamilies ( > 90% global pairwise nucleotide identity within families) (Supplementary Fig. 15 and Supplementary Data 3). Up to seven subfamilies could be detected in one species (*V. cholerae*) (Supplementary Fig. 15). In most cases, the distribution of the subfamilies coincides with the host species phylogeny. This is expected for mobile genetic elements transduced by vibriophages that, for the vast majority, have a narrow host range[22,23,35]. However, PICMI₁₁₅ was detected in two strains of *V. chagasii* (V115 and V157) isolated during the same time series sampling in France[22] and in a *V. toranzoniae* strain isolated from cultured clams in NW Spain (cmf 13.9) (Fig. 5 and Supplementary Fig. 16). Another *V. toranzoniae* isolate from seawater in SE Spain (96-373) carries a different PICMI subfamily. This incongruence between the subfamilies of PICMI and the bacterial hosts suggests that this satellite might be horizontally transferred between diverse *Vibrio* species.

We then analyzed the most frequent genes in the PICMI variants. As for PICMI₁₁₅, large non-coding regions often contained small ORFs

and/or pseudogenes (Supplementary Fig. 17). These non-functional genetic elements were not further considered in our analysis. The core genes encoding the integrase, the AlpA regulator and the primase were identified in all the elements, as expected, as they were used to identify the PICMI. We showed above that a gene of unknown function overlaps

*int* by four nucleotides (*iolg*) and is essential for the excision of the PICMI₁₁₅. Among the 35 PICMI subfamilies, 21 carry an *iolg* overlapping *int* over 1, 4, or 13 nucleotides, ATGA being the most frequent overlapping site ($n = 14$; Fig. 5 and Supplementary Fig. 15). Four elements carry a gene contiguous to the *int* gene (*icg*) and three do not carry a gene between *int* and *alpA*. The *iolg or icg* were grouped in 13 distinct gene families (>20% protein identity, 80% coverage), all of them of unknown function. The remaining seven PICMI elements were more divergent in gene repertoires and gene order. In six of them, *alpA* was in two or three copies. These loci may thus have been the result of multiple events of integration, gene loss, and recombination with other satellites. *Up1* homologs were found in 12 subfamilies of PICMI (Fig. 5 and Supplementary Fig. 15), in line with our observations that *alpA*, *up1*, and *prim* form the early regulon activated by the helper phage (Fig. 4). In 26 other PICMI subfamilies, single genes were also present between *alpA* and *prim*, forming eight distinct families, each encoding an unknown function. On the nine remaining subfamilies, *alpA* was adjacent to *prim*. Altogether, our analysis revealed that PICMI-like elements have small genomes and are widely distributed in the *Vibrionaceae*. Within their limited gene repertoire, a substantial proportion is demonstrated to be essential for the phage's lifestyle.

**Table 1 | PICMI₁₁₅ transfer by Φ115 phage produced from different donors**

| Donor strain | PICMI₁₁₅ titer[a] |
|---|---|
| PICMI₁₁₅-*Cm^R* | 6.13E + 03 ± 1.86E + 03 |
| Δ*int*-*Cm^R* | BDL |
| Δ*iolg*-*Cm^R* | BDL |
| Δ*alpA*-*Cm^R* | BDL |
| Δ*up1*-*Cm^R* | 5.37E + 03 ± 194E + 03 |
| Δ*prim*-*Cm^R* | BDL |
| Δ*up2*-*Cm^R* | 5.17E + 03 ± 172E + 03 |

*BDL* below the detection limit.

[a]PICMI₁₁₅ titer/ml of lysate, using *Vibrio* strain ΔPICMI₁₁₅ as recipient strain. The titer of phages produced from the different donors was 10⁶ PFU ml⁻¹. The average and standard deviations from three independent experiments are presented.

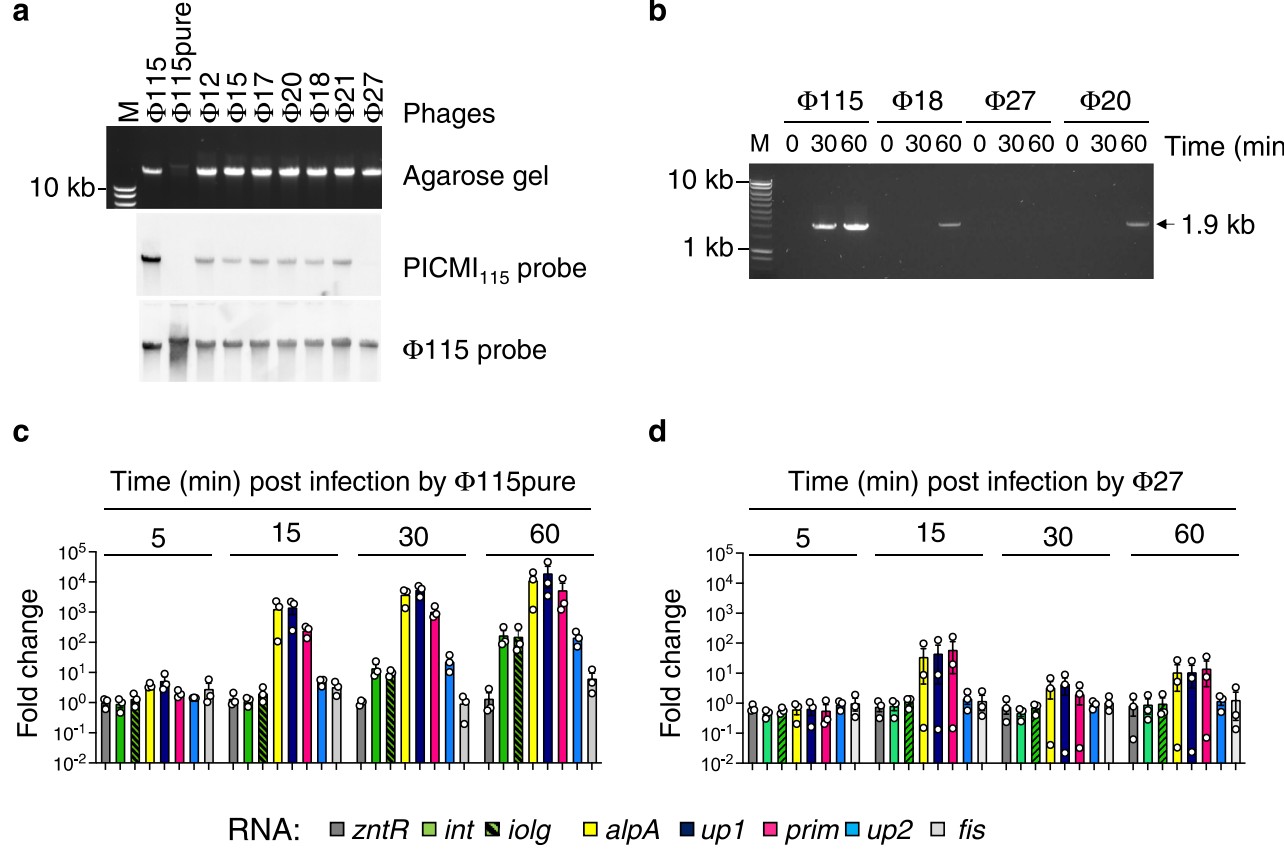

**Fig. 4 | The activation of PICMI₁₁₅ is helper phage specific. a** PICMI₁₁₅ DNA was less or not detected in Φ115 genetically related phages. DNA from viral particles were extracted and separated on SYBR Green stained gel and Southern blotted with an PICMI₁₁₅ or Φ115 probes. M: molecular marker (Smart ladder Eugentec). Uncropped gel and blots are provided as a Source Data file. Images are representative of two independent experiments. **b** An efficient induction of the satellite is specific to the helper phage Φ115. The *Vibrio* strain V115 carrying a Cm^R marked PICMI₁₁₅ was infected with the diverse phages at a MOI of 10 for the indicated time, PCR amplicon corresponding to the circular/concatemeric PICMI₁₁₅ were visualized on agarose gel. **c** The *alpA, up1,* and *prim* genes of PICMI₁₁₅ were observed to be upregulated early after infection with Φ115. In the experimental setup, *Vibrio* strain V115 was subjected to infection with either the pure Φ115 (**c**) or Φ27 (**d**). Utilizing qRT-PCR, the expression levels of the six genes from PICMI₁₁₅, as well as the flanking genes *fis* and *zntR*, and the housekeeping gene *gyrA* were assessed. Source data are provided as a Source Data file. Copy numbers were standardized to *gyrA* for each sample, and the fold change was determined by comparing samples collected at specified time points to the input samples taken immediately before introducing the phage. Bar charts show the mean ± SEM fold change from three independent experiments Individual data points from each experiment are represented by dots. At 15 min post-infection, one-tailed *t*-test result shows that only *alpA, iolg,* and *prim* induction by Φ115 result in a fold change >1 ($p < 0.05$).

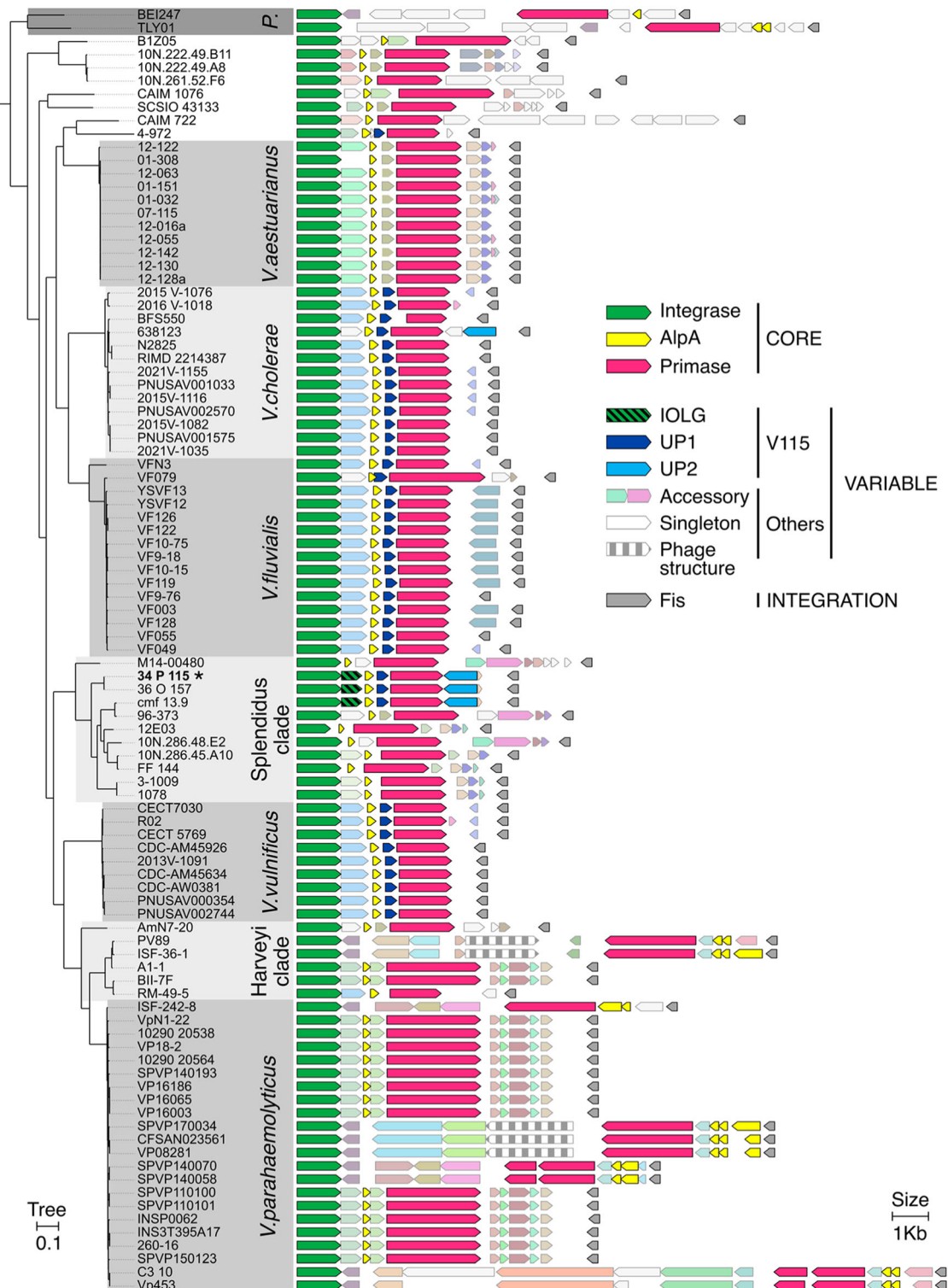

**Fig. 5 | PICMI-like satellite distribution and gene content in the *Vibrionaceae*.** Phylogenetic persistent core tree and genomic representation of the 97 PICMI elements found in *Vibrionaceae* (GenBank 01-27-23 containing 19189 organisms). Genus, super clades or of species names are indicated in the gray boxes. "P." corresponds to *Photobacterium* genus, Harveyi, Splendidus are super clades encompassing several *Vibrio* species. The PICMI₁₁₅ element is pinpointed by bold strain name (34_P_115) and by an asterisk. Solid colors indicate core PICMI₁₁₅ genes. Gray colors indicate accessory and singleton PICMI-like genes defined using reciprocal best-hit with 20% identity for 50% coverage. See also Supplementary Data 3 for details.

## Identification of a new defense system in PICMI₁₁₅

In spite of the small size of PICMIs, all subfamilies have accessory genes. Some of them encode genes with weak homology (<35%) to known virulence factors, whilst others encode for known phage defense systems (Supplementary Data 3). The latter are restriction modification systems type I and II, a retron type II, DRT_class_II, and Paris type I[36]. These anti-phage defense systems are located in the locus between *prim* and *fis*, suggesting that this might be a hotspot for the acquisition of anti-viral defense genes, akin to the locus between the integrase and Psu in P4-like satellites[20]. Although almost all the

elements where these systems were found lack homologs to *up1* and *up2*, this suggests that PICMI-like element can provide viral defense mechanisms to their host, as observed both in P4-like satellites and in PICI[21]. Consistent with this, we observed that the presence of PICMI₁₁₅ in V511 (transductants) greatly affected the infection outcome of this bacteria by the phage Φ511(Fig. 6a), showing an antiviral effect of the PICMI₁₁₅.

Since no known defense systems were detected specifically in PICMI₁₁₅, we hypothesized that the immunity provided by this element is mediated at least in part by a novel defense system, localized between *prim* and *fis* genes and encoded by the gene *up2*. To test this hypothesis, we cloned *up2* under the control of its native promoter in a plasmid and transferred it through conjugation to 46 other *V. chagasii* strains that are susceptible to at least one phage[22]. As a control, the same plasmid expressing *gfp* was transferred to the strains. Among the 90 possible host and phage combinations that led to the production of phage progeny, eight combinations, involving eight different phages, were affected by UP2 (Fig. 6b and Supplementary Fig. 18a). Six out of the eight phages affected by UP2 belong to the same family (Supplementary Fig. 18b), as defined by Virus Intergenomic Distance Calculator (VIRIDIC[37]) with pairwise identities ranging from 55 to 70%. Within this phage family, UP2 had no impact on the helper phage Φ115, whereas the titers of the remaining phages were reduced, ranging from partial (i.e. Φ120) to nearly complete abrogation (i.e. Φ511; Fig. 6c). The observed variation in UP2-mitigated infection suggests that coevolution may have already influenced this mechanism. This influence could manifest either through conferring an escape mechanism on the phage side or by triggering additional defense systems on the host side.

*V. chagasii* strain V511 was susceptible to a member of this VIRIDIC family, Φ511 (Fig. 6c) and, to a minor extent, to genetically more diverse phages Φ168 and Φ177 (Supplementary Fig. 18a). We found that UP2 influences the production of all three phages infecting the strain V511. UP2 anti-viral activity seemed, however, dependent on the V511 genetic background, as it was not observed for the combination involving phages Φ177 and Φ168 and other hosts (Fig. 6b), including the "original host" that was used to isolate the phage. Notably, the impact of the complete PICMI₁₁₅ element on V511 infection by Φ511 is significantly more pronounced compared to the presence of UP2 alone. The phage titer was reduced by $10^{-5}$ for PICMI₁₁₅ transconjuguants in comparison to the wild type, while it was $2.10^{-2}$ for V511 expressing only UP2 versus GFP (Fig. 6a, b and Supplementary Fig. 19). This implies that genes extending beyond UP2 contribute to the defense against Φ511. It is also conceivable that the activation/replication of the element interferes with Φ511 production, in contrast to Φ115.

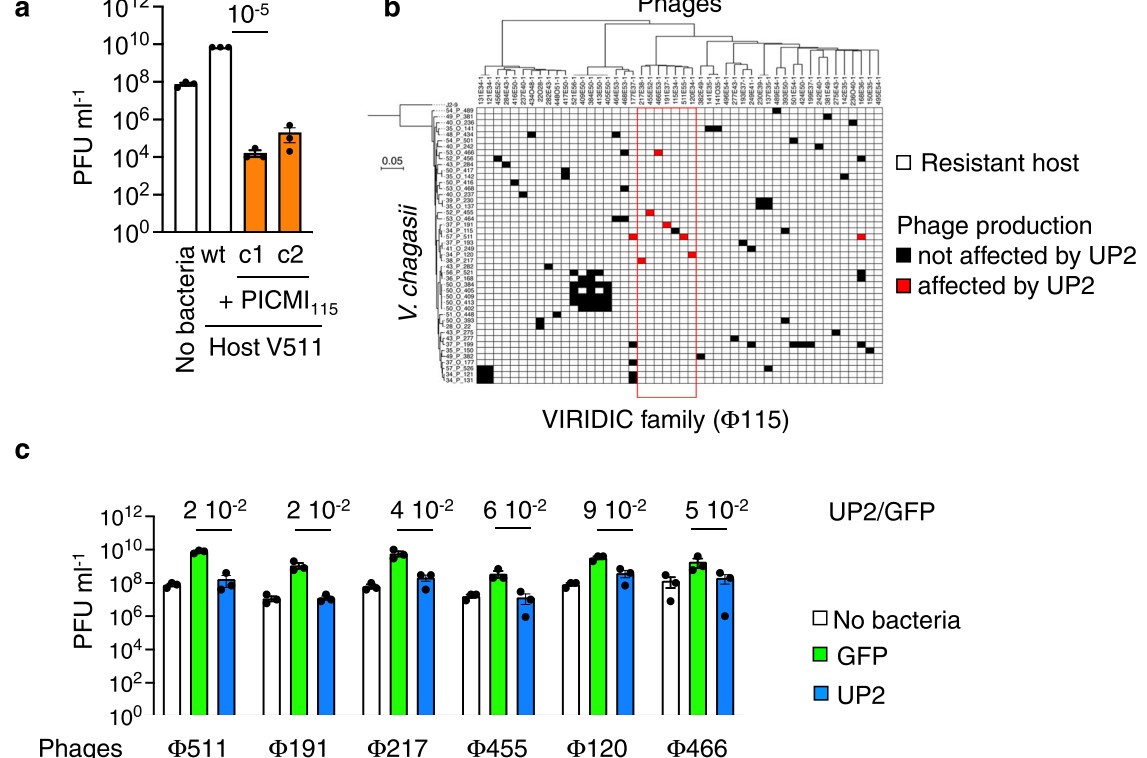

**Fig. 6 | PICMI₁₁₅ and UP2 confer host immunity to specific phages. a** PICMI₁₁₅ greatly affected the infection outcome by the phage Φ511. *Vibrio* strain V511 (wt) and transductants carrying the full satellite marked by Cm^R (PICMI₁₁₅, two clones c1 and c2) were infected with Φ511 at an MOI 10. The bar charts depict the average phage titer with the SEM from three independent experiments, represented by individual dots. The measurements were taken 60 min post-infection of the specified strains or for the equivalent volume of phage added to the culture media without bacteria. The ratio of phage titer in PICMI₁₁₅+ to PICMI₁₁₅- strain V511 is noted above the bar chart. **b** The gene *up2* encodes a novel defense system which activity depends on the phage and host genetic background. A plasmid carrying the gene *up2* under the control of its native promoter or, as control, the *gfp* under the constitutive promoter $P_{LAC}$, were transferred to diverse *V. chagasii* strains that are susceptible to at least one phage[22]. Rows represent sequenced *Vibrio* strains ordered according to the Maximum Likelihood persistent genome phylogeny of *V. chagasii* ($n = 46$). Columns represent phages ($n = 48$) ordered by VIRIDIC clustering dendrogram. Changes in susceptibility related to the presence of UP2 are indicated by a red square (Supplementary Fig. 18). **c** The susceptibility to UP2 of six phages that belong to the same VIRIDIC family than Φ115 was tested using their original host, i.e., the host used to isolate these phages. Phage titer was determined 60 min post-infection of the host or for the equivalent volume of phage added to the culture media without bacteria. The bar charts display the average phage titer with the SEM from three independent experiments, indicated by individual dots. The ratio of phage titer in UP2 to GFP strains is provided above the bar chart. A one-tailed t-student tests with Log10 transformation reveal that UP2 < GFP with *p*-values of 0.0112 (Φ511), 0.0004 (Φ191), 0.0091 (Φ217), 0.0250 (Φ455), 0.0294 (Φ120), and 0.1024 (Φ466).

We conclude that PICMI protects the bacterial host from non-helper phages. This protection relies at least in part on a novel UP2 defense system, whose activity is dependent on the host and phage genetic background. This highlights that certain defense systems exhibit such specificity that they can only be effectively studied within a limited genetic context, encompassing both the phage and its host. This underscores the importance of utilizing collections derived from natural populations and cross-infection matrices to enhance the relevance and applicability of findings.

## Discussion

Here, we report a new family of phage satellites packaged as concatemers in viral particles. The multiplicity of the concatemers is such that the packaged molecule of DNA has the size of the helper phage. As a result, PICMI does not require gene(s) involved in re-shaping the capsid size. The PICMI family is among the smallest of phage satellites, with PICMI$_{115}$ being the smallest satellite element with characterized activity. At the other end of the spectrum, cf-PICI[11] produce their own capsids dedicated to the exclusive packaging of their genome. PICMI minimalist gene repertoire seems dedicated to genes for excision and integration, DNA replication, and anti-viral defenses toward competitors of the helper phage. The small size of the PICMI implies a high dependency on the helper phage for activation, packaging, and release of the particles in the bacterial lysate. In our model system of PICMI$_{115}$, bacterial strain V115 and phage Φ115, this dependency seems costless for the helper phage. This observation aligns with earlier results for cf-PICI, in which production of the helper phages were also not significantly altered in the presence of a functional phage satellite[11].

The discovery of the PICMI family, and more specifically the mechanistic characterization of the PICMI$_{115}$ life cycle, is consistent with the previous hypothesis that capsid size reduction is not common among marine satellites[7]. Furthermore, PICMI lacks identifiable packaging genes, suggesting that it does not affect the composition of the viral particle, beyond packaging it with its own DNA. What are the advantages of packaging multiple copies of the satellite within native full-sized helper phage capsids? First, non-remodeled capsids might guarantee optimal interactions with the helper phage tail. Second, it diminishes the number of functions that must be encoded by the PICMI genome. Third, having multiple copies of the satellite injected by the viral-like particle into the cell could increase the expression of satellite genes that are necessary for its integration (gene dosage), thus increasing the frequency of integration after transduction. Finally, the acquisition of multiple copies of the extra-chromosomal element in one event could reduce the efficiency of host defenses. If larger satellites are packaged at a lower copy number, then there might be a trade-off between the acquisition of accessory genes and the efficiency of satellite transduction. This could explain why the genes present in the *prim-fis* hotspot region are restricted in number and subject to high turnover. A higher efficiency of transduction by polyploidization underscores a feature that makes PICMI unique. Indeed, PLE's transduction is severely reduced when packaged into ICP1-size capsids as concatemer or 6–7 PLE genomes relative to small remodeled capsids[9]. Future work will be needed to determine how the PICMI element is integrated or maintained as a single copy in the genome as we never detected more than one PICMI-like element in *Vibrionaceae* genomes.

PICMI induction requires cell infection by its helper phage. Early after infection, the regulon encoding *alpA*, *up1*, and *prim* is activated. The role of *up1* is unknown, but it is noteworthy that *up1* orthologs were found in 12 out of 35 subfamilies of PICMI and distributed in diverse *Vibrio* species (*V. cholerae, V. fluvialis, V. vulnificus,* and *V. chagasii*) (Supplementary Fig. 15). While *up1* is not essential for PICMI activation and spread, its presence in all elements of these subfamilies suggests it encodes for a trait under strong selection. *Prim*, which is necessary for efficient replication of the satellite encodes a putative RNA polymerase that synthesizes short fragments of RNA, which could be used as primers by the DNA polymerase. Homologs of the *prim* gene are also found in the vast majority of known satellites, in accordance with a core and essential function of this protein in the lifestyle of satellites[12]. AlpA appears as the key regulator of the switch from latency (integrated) to activation (excised) of PICMI. Indeed, its expression is necessary and sufficient to trigger the activation of the satellite in the absence of the helper phage. Our results suggest that the PICMI *int* gene is constitutively expressed and that phage-induced *alpA* is required for the formation of a functional excision complex. This leaves two relevant questions to be answered. How does the helper phage activate the early regulon? How do AlpA, the integrase, and probably IOLG interact to catalyze the excision of the satellite?

After PLE, PICMI is only the second family of satellites whose known helper phages are virulent. In contrast to PLE, PICMI does not significantly affect the production of its helper phage. We showed that PICMI can confer immunity to other virulent phages, and we identified a new defense system (UP2) encoded by the satellite. The phage range of UP2 activity appears very narrow, and many phages susceptible to this system are phylogenetically related to the helper phages. Hence, by protecting the bacteria if the phage is not a helper, PICMI is also protecting itself. UP2 promotes the stable coexistence of both helper phage and satellites within the bacterial populations.

Satellite, helper phage, and bacterial host interactions are highly specific. With such a narrow host range, how do the right combinations of phages, satellites, and bacterial hosts can meet in the marine environment? We speculate that blooms of specific *Vibrio* strains can dramatically increase the abundances of specific phages and satellites. When colonizing an animal host such as an oyster, the vibrios can reach a high density that favors physical contact and promotes phage infection and satellite transduction. The distribution of satellites in the environment is expected to adhere closely to the distribution of the helper phages at the short temporal and spatial scales due to the dependency of the former on the latter.

We recently highlighted that many phage defense genes are encoded on large genomic islands, named phage defense elements, but the mechanisms of transfer of these elements remained unexplored[23,30]. Around 0.6% of marine viral particles ($3.2 \times 10^{26}$ globally) are packaged satellites[7], and the discovery of PICMI-mediated immunity strongly suggests that phages, including virulent ones, play an important role in the mobility of the phage defense elements. A common view is that only virulent phages should be used for phage therapy to limit horizontal gene transfer. However, our data suggest that this idea should be taken with caution because PICMI$_{115}$ was efficiently transduced by a virulent phage. Indeed, the discovery of PICMI$_{115}$ and its helper virulent phage underscores the importance of understanding the interactions between virulent phages and the mobile genetic elements encoded by their bacterial hosts.

## Methods

### Bacterial strains and growth conditions

Phages and bacterial strains used in this study are listed in Supplementary Data 4 and 5, respectively. Strains used or established for the genetic approach are presented in Supplementary Data 6. *V. chagasii* isolates were grown in Marine Agar (MA, Difco) or Marine Broth (MB) at RT with gentle agitation. *Escherichia coli* strains were grown at 37 °C in Lysogeny Broth (LB, Difco) agar or in LB broth with shaking. Chloramphenicol (Cm; 5 or 25 µg ml⁻¹ for *V. chagasii* and *E. coli*, respectively), thymidine (0.3 mM) and diaminopimelate (0.3 mM) were added as supplements when necessary (all chemicals from Sigma-Aldrich). Induction of the P$_{BAD}$ promoter was achieved by the addition of 0.2% L-arabinose to the growth media, and conversely, was repressed by the addition of 1% D-glucose. Conjugation between *E. coli* and vibrios were performed at 30 °C as described previously[38] with the exception that we used TSA-2 (Tryptic Soy Agar, Difco, supplemented

with 1.5% NaCl) instead of LB for mating and selection. Briefly, overnight cultures of donor and recipient were diluted at 1:100 in culture media without antibiotic and grown up an $OD_{600nm}$ of 0.3. The mating was performed on TSA-dap using a donor/recipient ratio of 5/1. Counter-selection of ΔdapA donor was done by plating on a TSA devoid of diaminopimelic acid and supplemented with antibiotic.

## Phage isolation, high titer stock, and titration

New phages infecting V115 were isolated from concentrated seawater viruses sampled in summer 2021 in the same oyster farm and using the same protocol than in[22,23]. A volume of 100 μl of an overnight (ON) culture of bacterial host and 20 μl of viruses were directly plating on a bottom agar plate (1.5% agar, in MB) and 3.5 ml molten top agar (55 °C, 0.4% agar, in MB) were added to form host lawns in overlay and allow for plaque formation[39]. Plaque plugs were first eluted in 500 μl of MB for 24 hours at 4 °C, 0.2-μm filtered to remove bacteria, and re-isolated twice on V115 for purification before storage at 4 °C and, after supplementation of 25% glycerol at -80 °C. High titer stocks (>$10^{11}$ PFU ml$^{-1}$) were generated by confluent lysis in agar overlays[39]. To determine the titer of phage, bacterial lawns were prepared by mixing 100 μl of on overnight culture of cells with top agar and poured onto plates. Then, tenfold dilutions of phage were spotted on plate, which were incubated at RT for 24 h.

## Plasmid construction

The primers (Integrated DNA Technology) and plasmids used or established in this study are listed in Supplementary Data 7 and 8 respectively. For the preparation of qPCR standards, each amplicon was PCR amplified using the RedTaq polymerase (VWR) and cloned in the plasmid pCR2.1 using the TOPO-TA Cloning™ Kit (Invitrogen).

For the mutagenesis of vibrios, cloning was performed using Herculase II fusion DNA polymerase (Agilent) for PCR amplification and the Gibson Assembly Master Mix (New England Biolabs, NEB) for insert-plasmid assembly, according to the manufacturer instructions. All cloning was confirmed by digesting plasmid minipreps with specific restriction enzymes and/or sequencing (Eurogentec).

## Nucleic acid extraction, amplification, and Southern blot

Prior to DNA extraction, phage suspensions (5 ml, >$10^{11}$ PFU ml$^{-1}$) were concentrated to approximately 500 μl on centrifugal filtration devices (30 kDa Millipore Ultra Centrifugal Filter, Ultracel UFC903024) and washed with 1/100 MB to decrease salt concentration. Alternatively, phages were concentrated using PEG 8000 1X and NaCl 1 M, incubated ON at 4 °C, centrifuged 30 min at 2,800 g, and the pellet was resuspended in 500 μl SM buffer (NaCl 100 mM, MgSO$_4$. 7H$_2$O 8 mM, Tris-Cl 50 mM). The concentrated phages were next treated for 30 min at 37 °C with 10 μl of DNAse (Promega) and 2.5 μl of RNAse (Macherey-Nagel) at 1000 unit and 3.5 mg ml$^{-1}$, respectively. The nucleases were inactivated by adding EDTA (20 mM, pH 8). DNA extraction encompassed a first step of protein lysis (0.02 M EDTA pH 8.0, 100 μg ml$^{-1}$ proteinase K, 0.5% sodium dodecyl sulfate) for 30 min incubation at 55 °C, a phenol chloroform extraction, and an ethanol precipitation. Bacterial DNA was extracted using the Wizard Genomic DNA Purification Kit (Promega).

RNA was extracted with TRIzol™ Reagent (Sigma-Aldrich) and High Pure RNA Isolation Kit (Roche), treated by TURBO DNAse (Ambion), and reverse transcribed using the Transcriptor First Strand cDNA Synthesis Kit (Roche).

Classical PCRs were performed using the RedTaq (WVR) and amplicons were visualized by SYBR Green stained (Sigma) agarose gel electrophoresis (1 to 2% agarose). qPCR and qRT-PCR was performed using LightCycler 480 SYBR Green I Master (Roche). The thermal cycling conditions were 95 °C for 10 min, followed by 40 cycles of 95 °C for 10 s, 60 °C for 20 s, and 72 °C for 25 s, then 1 cycle of 95 °C for 5 s, 65 °C for 1 min and 95 °C for 15 s. Standard curves were constructed using serial dilutions of plasmid, leading to the number of DNA copies per 20 ng of DNA. For all assays, three independent DNA samples (i.e., biological replicates) were tested under each condition. The number of copies for the phage, the empty integration site, and the circular/concatemeric PICMI115 was normalized by the number of copies of bacteria (gyrB) per sample. In the qRT-PCR analysis, the resulting copy numbers were further normalized to gyrA for each sample. Fold change was determined by comparing samples collected at specified time points to the input sampled immediately before adding the phage.

For Southern blot, DNA samples were run on 0.7% agarose gel at 100 V for one hour. Then, the DNA was transferred to Nylon membranes (Hybond-N+; Amersham Life Science) using standard methods. DNA was detected using a DIG-labelled probe (Digoxigenin-11-dUTP alkali-labile), anti-DIG antibody (Anti-Digoxigenin-AP Fab fragments) and Chemiluminescent detection with CSPD following the instructions of the kit (all products and kits from Roche).

## Construction of HiC libraries, sequencing, and analysis

1 ml of different mix of high titer stocks (>$10^{11}$ PFU ml$^{-1}$) of phages (mix1: 1 ml of Φ115, mix2: 1 ml of Φ191, mix3: 500 μl of Φ115 + 500 μl of Φ191) were fixed in a 5 ml Eppendorf tube by adding formaldehyde (Sigma-Aldrich, ref - F4775, Formalin 35-36.5% plus methanol 15%) to a final concentration of 3% and incubated at RT for 1 hour under gentle agitation. The reaction was stopped by adding glycine (stock = 2.5 M) to a final concentration 0.125 M and incubated at RT for 20 min under gentle agitation. Fixed particles were then centrifuged at 16,000 g for 20 min at 4 °C. Supernatant was discarded, resuspended in 1 ml of PBS 1X, and recentrifuged at 16,000 × g for 20 min at 4 °C. The supernatant was again discarded carefully and the pellet was resuspended in 45 μl of Tris 10 mM pH 7.5. The HiC libraries were then constructed using the ARIMA Kit (Arima Genome-Wide HiC+ Kit). HiC genomic libraries were then processed for sequencing as previously described[40] and were sequenced on Nextseq550 apparatus (2 × 35 bp). Contact maps were generated using Hicstuff[41] (bowtie2 - very sensitive local mode – mapping quality of 30) and a reference FASTA files containing the 3 phage genomes. Contact maps were then binned at 1 kb resolution, balanced, and displayed using Hicstuff.

## Electron microscopy

Following concentration on centrifugal filtration devices (Millipore, amicon Ultra centrifugal filter, Ultracel 30 K, UFC903024), 20 μl of the phage concentrate were adsorbed for 10 min to a formvar film on a carbon-coated 300 mesh copper grid (FF-300 Cu formvar square mesh Cu, delta microscopy). The adsorbed samples were negatively contrasted with 2% Uranyl acetate (EMS, Hatfield, PA, USA). Imaging was performed using a Jeol JEM-1400 Transmission Electron Microscope equipped with an Orious Gatan camera at the platform MERIMAGE (Station Biologique, Roscoff, France).

## Vibrio mutagenesis

PICMI labeling (PICMI$_{115}$-Cm$^R$) was performed by cloning the 500 bp end of up2 gene in the suicide plasmid pSW23T[42]. For the mutagenesis, two approaches were employed, depending on the target gene (Supplementary Fig. 20). Firstly, for gene deletion, 500 bp fragments flanking the gene were cloned into the pSW7848T suicide plasmid[43]. This vector encodes the ccdB toxin gene under the control of an arabinose-inducible and glucose-repressible promoter, P$_{BAD}$[38]. Selection of the plasmid-borne drug marker on Cm and glucose resulted from integration of pSW7848T in the genome. The second recombination leading to pSW7848T elimination was selected on arabinose-containing media. Mutants were screened by PCR using external primers. For transduction experiments, mutants were marked by pSW23T insertion at the up2 end, as previously described. Secondly, for up2 and prim, a 500 bp internal region of

the gene was cloned into the suicide plasmid pSW23T (Supplementary Fig. 20). After conjugative transfer, plasmid-borne drug marker selection (CmR) resulted from the integration of pSW23T in the target region through a single crossing-over, leading to simultaneous gene inactivation and PICMI labeling. Integration of the suicide plasmid was confirmed by PCR using one primer in the plasmid and one in PICMI.

For the complementation experiments, the genes necessary for PICMI$_{115}$ excision, int/iolg and alpA, or a gfp control were cloned under the control of the conditional $P_{BAD}$ promoter in a P15A-ori-based replicative vector. The plasmids were transferred by conjugation into the mutants. Strains were grown to mid-exponential phase in the presence of 0.2% arabinose (activation of $P_{BAD}$) and then infected with Φ115pure for 30 min. To explore its anti-phage activity, up2 under its native promoter was cloned in a pMRB plasmid[44], and the same plasmid expressing the gfp was used as control.

We were not able to delete the complete PICMI115 by allelic exchange using the pSW7848T suicide plasmid. As an alternative, we cloned the alpA gene of the V. chagasii or V. aestuarianus under the control of $P_{BAD}$ promoter in a P15A-ori-based replicative vector (Spec$^R$), assuming that the expression of alpA in trans is sufficient to induce PICMI$_{115}$ excision even in the absence of phage (Fig. 3c) and the alpA from the two Vibrio species are interchangeable for PICMI$_{115}$ induction (Supplementary Data 9). The plasmid was transferred by conjugation to V115. The Spec$^R$ conjugant was grown overnight in MB with Spec and arabinose, serially diluted and plated on TSA-2 (TSB-2 with agar). A total of 48 colonies were screened by PCR to identify V115 derivatives that lack the PICMI (ΔPICMI). We obtained clones with the mutation (8/48) only when using alpA from V. aestuarianus.

## PICMI induction
The Vibrio strain was grown to mid-exponential phase in MB (OD = 0.3) and infected under static condition with the phage at a MOI of 10 otherwise indicated. At each time point, an aliquot of the culture was centrifuged, the supernatant was filtered at 0.2 μm and the titer of phages was determined by drop spotting serial dilutions of the supernatant on the host lawn. Total RNA and/or DNA was extracted from the bacterial pellet.

## Adsorption estimation
Phage adsorption experiments were performed as previously described[45]. Phages were mixed with exponentially growing cells (OD 0.3; 10$^7$ CFU ml$^{-1}$) at a MOI of 0.01 and incubated at RT without agitation. At different time points, 250 μl of the culture was transferred in a 1.5 ml tube containing 50 μl of chloroform and centrifuged at 15,000 g for 5 min. The supernatant was 10-fold serially diluted and drop spotted onto a fresh lawn of a sensitive host to quantify the remaining free phage particles. In this assay, a drop in the number of infectious particles at 15 or 30 min indicated bacteriophage adsorption.

## PICMI transduction
The number of PICMI particles were quantified using the transduction titering assay. Briefly, lysates were produced by infecting V115 derivatives carrying PICMI$_{115}$-Cm$^R$ and derivatives by Φ115pure. A 1:100 dilution (in fresh MB broth) of an overnight recipient strain was grown until an OD$_{600}$ of 0.3 was reached. Then, 100 μl of the recipient culture was dispatched in a 96-well plate, infected by addition of 10 μl of PICMI lysate serial dilutions prepared with MB for 1H at RT. The different mixtures of culture-PICMI-Cm$^R$ were plated out on TSA-2 plates containing chloramphenicol. LBA plates were incubated at RT for 24 h, and the number of colonies formed (transduction particles present in the lysate) were counted and represented as the colony-forming units (CFU ml$^{-1}$). PCRs were performed to confirm the integration of PICMI at the end of the fis gene.

## In silico prediction and analysis of PICMI-like element
The PICMI-like elements were searched using two datasets: (1) the bacterial division of GenBank release v243 (5/26/2021) that contains 24,243 complete genomes, including 456 genomes of the Vibrionaceae family and (2) the NCBI Assembly database (2/16/2023) with 19,185 Vibrionaceae genomes available but with variable assembly quality. SatelliteFinder (v0.9.1)[12] was used on both datasets with a dedicated PICMI model (Supplementary Software) defined by four mandatory genes encoding: the integrase (PF00589.25, PF00239.24, PF07508.16), AlpA (PF05930.15, PF12728.10), the primase (DUF3987 with PF13148.9 and DUF5906 with PF19263.2) and the Fis regulator (PF02954.22). The resulting elements were then filtered by excluding those with int, alpA, and fis localized in different contigs, those predicted to belong to other families (PICI, cf-PICI, P4, and PLE), and those integrated into a gene showing a lower identity with fis. The genomic region starting with the fis gene and ending with the direct repeat upstream the int gene was extracted, aligned with FAMSA (v1.6.2)[46], and each PICMI subfamily was defined by a pairwise nucleic identity ≥90% (Supplementary Data 3 and Fig. 15).

The PICMI genes were clustered in families using mmseqs2 (v14.7e284) reciprocal best-hit[47] with 20% identity and 50% coverage thresholds (Supplementary Data 3 and Figure 15). Phage defense systems were annotated using Defense-Finder (v1.2.0 and models v1.2.3)[36] and phage structural genes were annotated using diamond blastp (v2.0.8) on PhANNs database from 05/25/23[48] with a threshold score ≥7. Virulence factors in PICMI were searched using diamond blastp (v2.0.6) with the Virulence Factor Database proteins[49], and with the parameters --query-cover 50, --ultra-sensitive and –forward only. No antibiotic resistance genes were found in PICMI using AMRFinder Plus (v3.11.0)[50].

The functional annotation of genes used multiple approaches, i.e., tblastn similarity searches on GenBank, InterPro domain prediction[51], and ProtNLM annotation [https://www.uniprot.org/help/ProtNLM]. The 3D protein structures predicted by ColabFold (v1.5.2-patch)[33] were compared to publicly available protein structures using the Foldseek search server (v5)[52]. All data are provided as supplementary Data 10.

Comparative genomics were performed using PanACoTA workflow (v1.4.0)[53]. Persistent genes were defined as present in single copy in at least 90% genomes with a minimum of 30% protein identity. Protein sequences of each family were first aligned and concatenated. Phylogenetic reconstruction was done using iqtree2 (v2.0.3)[54] with 1000 bootstraps and GTR model. Genome plots were generated using dedicated python scripts based on the 'DNA Features Viewer' library (https://github.com/Edinburgh-Genome-Foundry/DnaFeaturesViewer).

We clustered phages using VIRIDIC (v1.1, default parameters)[37]. Intergenomic similarities were identified using BLASTN pairwise comparisons. Viruses' assignment into genera (≥ 70% similarities) and species (≥ 95% similarities) ranks follows the International Committee on Taxonomy of Viruses (ICTV) genome identity thresholds. We used PHANOTATE v1.5.0[26] for the syntactic annotation of phage Φ115 and, after removing restriction on gene size, analysis of large non-coding regions within PICMIs.

## Nanopore genome assembly and analysis
The Nanopore sequencing library was prepared using Native Barcoding genomic DNA (EXP-NBD104) and Ligation Sequencing Kit 1D (SQK-LSK109) and sequenced using MinION flow cell R9.4.1 at the platform GENOMER (Station Biologique, Roscoff, France).

Demultiplexing and base calling of raw nanopore sequencing data (Supplementary Data 1) was performed using Guppy software (v6.1.1, --flowcell FLO-MIN106 --kit SQK-LSK109). The base called sequences were used as input for genome assembly performed using FLYE (v2.9)[55] RAVEN (v1.4.0)[56] and NECAT (v0.0.1)[57] with default parameters.

The comparisons of Illumina and Nanopore assemblies of both Φ115 and PICMI$_{115}$ were performed using pairwise FAMSA alignment

(v1.6.2)[46]. FLYE was selected for further analysis because it showed the highest similarity with previous Illumina sequencing. Nanopore PICMI reads were analyzed using a dedicated script based on FAMSA alignment to precisely determine read start and end on an artificial 13 copies concatemer reference.

## Reporting summary

Further information on research design is available in the Nature Portfolio Reporting Summary linked to this article.

## Data availability

Accession numbers of phage and vibrio genomes isolated and sequenced in ref. 22 are listed in the Supplementary Data 4 and 5 respectively. Source data are provided with this paper.

## Code availability

The MacSyFinder models used to identify PICMI are provided (Supplementary Software) and can be used with the MacSyFinder to make novel analysis. MacSyFinder is available in a public repository at https://github.com/gem-pasteur/macsyfinder.

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

## Acknowledgements

We thank Jose Penades, Agnes Thierry, Céline Loot, Francois-Xavier Barre, and Mélanie Blokesch for valuable suggestions. We thank Sophie Le Panse (MERIMAGE, Roscoff), Gwen Tanguy, Erwan Legeay (GENOMER, Roscoff), Karine Cahier, and Yannick Labreuche for technical assistance. We thank Jenna Sternberg from Life Science Editors for help with the Manuscript. This work was supported by funding from the European Research Council (ERC) under the European Union's Horizon 2020 research and innovation program (grant agreement No 884988, Advanced ERC Dynamic) to FLR, from the Agence Nationale de la Recherche (ANR-20-CE35-0014 « RESISTE ») to EPCR and FLR. R.B.-C. acknowledges the Spanish Ministerio de Ciencia e Innovación for his FPI predoctoral contract (BES-2017-079730). EPCR lab was funded by Equipe FRM (Fondation pour la Recherche Médicale): EQU201903007835. This work used the computational and storage services (TARS cluster) provided by the IT department at Institut Pasteur, Paris.

## Author contributions

F.L.R. conceived the study, supervised the project, and secured funding. R.B.C., D.P., M.M. and F.L.R. conducted the experiments. D.G., J.M.S. and E.P.C.R. performed the genomic analyses. R.B.C., D.G., J.M.S., D.P., M.M., E.P.C.R. and F.L.R. analyzed the data. F.L.R., R.B.C., J.M.S. and E.P.C.R. wrote the manuscript.

## Competing interests

The authors declare no competing interests.
