## [Peer Review File · Nature Communications]

Phage-inducible chromosomal minimalist islands (PICMIs), a novel family of small marine satellites of virulent phagesReviewer #1 (Remarks to the Author):

The manuscript "Phage inducible chromosomal minimalist island (PICMI). A family of satellites of marine virulent phages" by Barcia-Cruz et al. describes the discovery and characterization of a novel type of phage satellite in *Vibrio* and other marine bacteria that in contrast to known satellites such as the PLEs or classical phage inducible chromosomal islands (PICIs) are very small and do not encode genes that can redirect phage packaging towards the satellite and also miss genes that could function in remodelling helper phage capsids. As such, they encode only a very limited set of core genes (integrase, and integrase overlapping gene, primase and *alpA*) that appear to be involved at various stages of the replication cycle. The authors propose a role for *AlpA* in the formation of the excision complex rather than a transcriptional regulator. This will need to be verified in future studies. The limited number of core genes also is reflected in an average size of PICMIs that is lower than that of other phage satellites. Given the shortness of the element, it is perhaps not surprising that functions such as phage remodelling and packaging redirection are not encoded within the PICMI and that there is no appreciable impact of PICMI mobilisation on helper phage propagation. Nevertheless, this highlights that diverse replication and exploitation strategies (of helper phages) can be evolutionarily successful. The authors also identify a cargo region that can contain a range of different auxiliary genes and identify that one of these in PICMI115 (*UP2*) confers protection of the resident strain to infection by non-helper phages, suggesting that these elements are beneficial to the host bacterium. Overall, the manuscript is very well written, and the core hypotheses are supported by the evidence provided. I think this is an important discovery and adds a key element to the diversity of phage satellites and their role/importance in the ecology of bacteria and their predators (particularly because so few have been identified to respond to virulent phages). I have no major comments regarding the manuscript.

General comments:

The authors focus on one genetic context/integration site, and it is possible that they do not capture the full breadth of PICMI elements as there might be other elements in different integration sites. How do the bioinformatically identified core genes that do not map to this site compare and is there any signature of PICMIs, etc to be found in other genomic locations?

Please take a careful look at the colour schemes used in different figures as for some it is hard to make out the difference between red and brown (i.e. Figure 1A red prim gene looks very similar to the "brown" *CmR* triangle. Please consider using colour-blind friendly palettes in figure preparation (<https://www.nature.com/articles/d41586-021-02696-z>).

Specific comments:

Line 64: provide a citation for average size ranges of satellites

Lines 113-114: what was the rationale for deciding that these small ORF were not genuine? Do they lack an RBS? Proteins of a similar size have previously been shown to be involved in cross-talk between MGEs as well as interference mechanisms. I would be careful in outrightly discarding short ORFs without additional justification.

Line 144: replace "has enabled" with "has allowed" to avoid repeating enabled.

Lines 164-165: The authors suggest that their data is in line with rolling-circle replication. However, they also see an increase in circularisation over time (Figure 1C), which would suggest that there might be an element of replication of the circular form of the PICMI. Is this correct or an artifact of data presentation? See also comment about figure below.

Lines 177-178: Not sure that this is copy numbers. It rather sounds like the number of phage particles carrying PICMIs increased for the PICMI115-*CmR* compared to the wt PICMI. The change is not insignificant (from 16% to 57% of the viral particle population) from Figure S6 and there now seems to be some level of interference with the phage that is being observed. For example, the phage titre now drops by almost 1-log unit compared to the titre observed when the wt PICMI strain was infected. PICMI titres from the marked PICMI seem to be similar to those obtained from the wt PICMI strain. This drop in phage titre disappears in either of the mutants that fail to induce the PICMI (indicating that it is related to PICMI induction) but is also absent on both *USP1* and *USP2* mutants. It seems unlikely that *USP1* is related to this phenotype since it is upstream of the *CmR* suicide plasmid insertion. However, *USP2* expression might be affected by the insertion of the plasmid, and this could lead to higher *USP2* levels resulting in defence against the helper phage.

Loss of USP2 restores phage titres. Note that in Figure 3C, USP2 is not induced as much as other genes by phage infection and the level of USP2 induction might well be what determines interference. Do you see this drop in copy numbers with the unmarked deletion mutants? Can you express different levels of USP2 prior to Phi115 infection to determine if this reduced the phage titre? Please note that I am not requesting this experiment for the manuscript to be accepted but rather as I found it an interesting observation.

Lines 181-185: The formulation is a bit unclear and too dense to easily follow it. I would suggest rewording it, so the two infection experiments are separated and described individually as clearly different MOIs were required depending on the host strain.

Lines 237-238: "This resulted from the increase of the number of copies of the PICMI115 genome and not necessarily through gene activation." Please provide/refer to evidence.

Line 330: "Their minimalist gene" replace "Their" with "PICMI" to specify/clarify

Line 337: Define what is meant by "insignificant". 50% can be quite significant biologically although it would not show up as statistically significant in phage titre data.

Line 447: "Φ115pure at MOI10 for 60 minutes" space missing between MOI and 10.

Line 453: Greek symbols missing "the phage F115pure".

Line 455-458: Greek symbols missing for phages and mutants.

Line 459: "ILOG" should not be capitalised to be consistent with figure and gene nomenclature normally using non-capitalised spelling.

Line 461: Replace "was detected by classical PCR and gel stained (upper panel)" with "was detected by classical PCR and separation on agarose gel (upper panel)"

Lines 463-477: Greek symbols missing for phages and mutants.

Line 482: Replace "correspond" with "corresponds".

Line 517: Remove "upon request and"

Line 520/561 (and throughout document): Capitalise "Vibrio" as a genus name.

Line 529: "Vibrio. Chagasii" either abbreviate or remove full stop. Check capitalisation.

Line 572: correct formatting of "MgSO₄.7H₂O"

Lines 589-591: Specify method for fold change calculation.

Lines 622-627: Please expand paragraph as the two approaches for generating mutants are not completely clear as described.

Line 623: change "and at the same time label the" to "and at the same time as labelling the"

Line 624: formatting of "Δprim and ΔUP2 -CmR", are both mutants CmR-marked?

Line 668: change "tittering" to "titring"

Line 672-674: A one hour incubation for a transduction seems very long considering that the phage replication cycle is less than that. Is there a particular reason for selecting this time frame and is it possible that transduction titres could be lower/higher because of repeated infection cycles during incubation?

Line 753: Vibrio cholerae formatting in reference

Figures:

Figure 1C: Unclear to what fold-change refers to. Is it to the values at 5 min, the average of the values at 5 min? Surely for excision and circularisation this should be 0 min to reflect baseline levels before infection. What is the phage baseline? I am struggling a bit with making sense of this data as it is presented. It might make more sense to present the excision data as percentage of the bacteria in which the PICMI has excised related to the total number of sites available would be the sum of those occupied (one of the grey + one of the black primers in figure 1A) or excised. Similarly, it is unclear whether the increase in circularisation is reflective of replication of circular forms of the PICMI or simply the result of different portions of the excised PICMI slowly circularising before entering rolling circle replication.

Figure 2C. Contrast of agarose gels too high to see whether individual gels or a single gel. I don't think the fold change adds anything here and because of the no-change for alpA, it might be more confusing than informative. Figure S11 is far more informative here.

Figure 5B: I am missing the effect size in this graph. How strong is the interference for each pair? Also, I cannot see the control plasmid on the graph. Is this a relative ratio between the control and the UP2 plasmid within the same strain background? Does no phage production mean that the phage was unable to infect either the strain containing UP2 or gfp? I would suggest modifying this figure to incorporate effect sizes into to, potentially colour coded by fold-change, etc.

Figure S6: Can you explain why you cans till detect PICMI-115 particles in the ΔPICMI115 mutant lysate? I would expect them to be not-detected in a clean lysate. Is there carryover, contamination

or a limitation of the assay that does not allow to detect below a certain level of copies? What is the limit of detection of the assay? Can this be indicated in the figure? Please also show the reference curves used for calibration of the copy numbers.

Figure S9: Define the bar to the left of the figure. Increasing dilutions/decreasing concentrations?

Reviewer #2 (Remarks to the Author):

Congratulations to the authors; this is a fascinating piece of work in which the authors have characterized a new family of satellites in Vibrionaceae, named the phage-inducible chromosomal minimalist Island (PICMI). The text is pleasant to read, I enjoyed following the rationale and logic of their experiments after their observations on the two contigs that appeared on their initial exploration on *V. chagasii* phages.

The manuscript provides a convincing argument as to why this family should be considered related to the PICMI umbrella yet kept apart with a different name due to their unique mutualistic lifestyle. The authors neatly show that PICMIs are capable of being induced and transferred by lytic phages, while altering very little the production of their inducing phage. Importantly, their distribution amongst Vibrionaceae species suggests they could greatly impact phage diversity and play an important role in horizontal gene transfer as other satellites do.

I would like to suggest a change in the title:

"Phage inducible chromosomal minimalist island (PICMI), a new family of small marine satellites of virulent phages."

I have only minor comments and suggestions that the authors should address:

General comment:

Figures with bar charts. Please use standard deviation and appropriate statistical test to validate significance. This is a standard form for the journal.

When reporting phage titer, I personally find it easier to read when the figure depicts PFU/ml and not fold change, as I get an overview from the initial phage titer and can be related with the transduction titer (TFU/ml). I would leave this to the editor and consider if this could help the reader.

PICMI-like elements distribution: These elements clearly tend to be minimal. Were there similar satellites found in other bacterial species?

Identification of defense system: I really like the authors' approach to identify and test such genetic modules contained within the PICMI. However, in some instances, the fold change reduction on phage titer is just a 100-fold and this works with few phages as seen in Figure 5. Would it be possible that UP2 mitigates infection in a different manner by controlling a specific set of genes in the host that cannot be used by such phages? I think the authors should provide more information as to why UP2 should be considered an anti-viral module and why this would exclusively protect from non-helper phages.

Discussion:

I would encourage the authors to make a comment regarding the packaging of the PICMI and how this new family can be a significant aspect to pathogenicity.

Could UP1 have a role in packaging?

PICMIs in pathogenic strains tend to carry toxins, antibiotic resistance and/or phage interference genes. Is there any evidence that PICMIs contribute to the virulence of their host? Are PICMIs associated with strains that can cause disease?

Can the newly characterized defense system UP2, in PICMI115 influence other MGEs, such as plasmids?

Comments:

Line 25 and whenever referred to phage-inducible chromosomal islands, keep the hyphen for consistency with other related works.

Line 50-52. I would like the authors to revise such paragraph as it lacks some information. PICIs are more abundant and spread in the bacteria phyla than other satellites, including species in Firmicutes, Enterobacteriaceae, Bacillaceae, and Gammaprotobacteria. Cf-PICIs are a bit less abundant and mainly in Enterobacteriaceae, Lactobacilli, Enterococci, Pasturella, Bacillaceae, Streptococci and Morganellaceae. P4s are mainly in Enterobacteriaceae, Yersiniaceae, Erwiniaceae, Hafniaceae and Pectobacteria.

Line 56. The authors explained some common features about satellite induction, but what about satellites induced by other satellites. Check Haag et al. (2021) Nat Microbiology.

Line 75. I share the views of the authors regarding the complex relation between bacteria, the phage, and the satellite. What about symbiosis? Could we start considering such elements as symbionts who play important role in evolution and adaptation?

Line 84. Change "we addressed this challenge..." to "we addressed these challenges..."

Line 109. Change "a putative regulator" to "a putative transcriptional regulator". Then this will make more sense when reaching the AlpA experiments later.

Line 114. Is there a reference for Phanotate?

In lines 144-150. This paragraph is exciting. I wonder if the authors could expand on a few aspects related to packaging. Typically, we found that the type of PICIs or satellites have similar packaging mechanism as their related helper phage (this being cos or pac). Is the inducing phage a cos or pac type? If the inducing phages have a TerS, can such protein be employed by the PICMI?

Line 171. I guess the authors could also verify and mentioned a few defense mechanisms using Padloc or DefenseFinder. Regarding fis, this is more curiosity, but what significance has fis and does it play a major function in the Vibro species?

Figure 1. fis is mentioned often in the text, but I have no background on its context. It would be a nice addition a small description of what fis and zntR are for its host, and if there is a relation with phage integration. Are there any phages integrated there?

Line 181-187 and Figure 1E. I appreciate the authors experimental design by normalizing the number of transductants by the number of phages, however this can be simplified by indicating the number of TFU/ml that were obtained.

Line 195-196 and Figure 1F. I would rather the authors to employ PFU/ml and remember to highlight what was the recipient or propagating strain.

Line 210. The leaky expression of alpA by the PBAD promoter seems to be the caused to this early excision, suggesting that the mechanism could be dose dependent.

Figure 2C.

The southern blot seems to indicate that prim deletion could have an impact on phage 115. Any comments as to why the band look so different from the rest? I would expect to have similar intensities for bands with int, OLG, alpA, prim and PICMI115 deleted if truly the PICMI does not impact phage reproduction.

I really like Figure S11. Maybe this one can substitute the lower section of Figure 2C, or be another panel as Figure 2D.

Please check the math symbols for this paragraph (Figure legend 2) as some have may not be transferred adequately.

Regarding AlpA; the authors compared its structure with Xis and TorL. However, I would like them to discuss the difference of this AlpA to the AlpA expressed by PICIs and other MGEs such as ICEs. Is this an annotation problem that should be addressed by the community employing better prediction software?

Lines 224-241. It seems as the seven phages are related to phage 115. Is phage 27 much different from the others? This could suggest what and how the PICMI has been induced. Possibly the differences between 27 and 115 could point towards the mechanism from which alpA is activated.

Lines 282-286. Any comments on what UP1 function could be?

Lines 313-318. Could there be other modules at work to hinder the interference? Maybe other elements within the host genome

Line 326. "These results..."

Line 350 "...this finding suggest a potential trade-off.."

Line 498 "...GFP strains are..."

Table S3 is missing from manuscript.

--I really enjoyed this story and will be looking forward to know more about PICMIs. RIC

Reviewer #1

The manuscript “Phage inducible chromosomal minimalist island (PICMI). A family of satellites of marine virulent phages” by Barcia-Cruz et al. describes the discovery and characterization of a novel type of phage satellite in *Vibrio* and other marine bacteria that in contrast to know satellites such as the PLEs or classical phage inducible chromosomal islands (PICIs) are very small and do not encode genes that can redirect phage packaging towards the satellite and also miss genes that could function in remodelling helper phage capsids. As such, the encode only a very limited set of core genes (integrase, and integrase overlapping gene, primase and alpA) that appear to be involved at various stages of the replication cycle. The authors propose a role for AlpA in the formation of the Excision complex rather than a transcriptional regulator. This will need to be verified in future studies. The limited number of core genes also is reflected in an average size of PICMIs that is lower than that of other phage satellites. Given the shortness of the element, it is perhaps not surprising that functions such as phage remodelling and packaging redirection are not encoded within the PICMI and that there is no appreciable impact of PICMI mobilisation on helper phage propagation. Nevertheless, this highlights that diverse replication and exploitation strategies (of helper phages) can be evolutionary successful. The authors also identify a cargo region that can contain a range of different auxiliary genes and identify that one of these in PICIMI115 (UP2) confers protection of the resident strain to infection by non-helper phages, suggesting that these elements are beneficial to the host bacterium.

Overall, the manuscript is very well written, and the core hypotheses are supported by the evidence provided. I think this is an important discovery and adds a key element to the diversity of phage satellites and their role/importance in the ecology of bacteria and their predators (particularly because so few have been identified to respond to virulent phages). I have no major comments regarding the manuscript.

General comments:

The authors focus on one genetic context/integration site, and it is possible that they do not capture the full breadth of PICMI elements as there might be other elements in different integrations sites. How do the bioinformatically identified core genes that do not map to this site compare and is there any signature of PICMIs, etc to be found in other genomic locations?

We look for PICMIs (their core genes) all over the genome. The integration site is defined by *fis* (which we define as a core gene for the identification of the element), but we find no variants that lack this component, i.e. there are no PICMI found at other integration sites. This data (the frequency of variants) have been included in the paper (Figure S14 below) and for the sake of future analysis by us and other researchers, we modified the SatelliteFinder model to include *fis* as an optional component.

Supplementary Figure 14. Number of the different variants of PICMI elements identified in bacterial genomes. Each circle represents a core component of the PICMI used in the SatelliteFinder model. The absence of a circle (in rows) corresponds to a PICMI variant where a particular component (for Type B) or two components (for Type C) is, or are, undetected.

Please take a careful look at the colour schemes used in different figures as for some it is hard to make out the difference between red and brown (i.e. Figure 1A red prim gene looks very similar to the “brown” CmR triangle. Please consider using colour-blind friendly palettes in figure preparation (<https://www.nature.com/articles/d41586-021-02696-z>).

We apology for this mistake. We used <https://www.color-blindness.com/coblis-color-blindness-simulator/> to improve the inclusivity of our Figures.

Specific comments:

Line 64: provide a citation for average size ranges of satellites

Done. Moura De Sousa et al., NAR 2023

Lines 113-114: what was the rationale for deciding that these small ORF were not genuine? Do they lack an RBS? Proteins of a similar size have previously been shown to be involved in cross-talk between MGEs as well as interference mechanisms. I would be careful in outrightly discarding short ORFs without additional justification.

We acknowledge the growing interest in small ORFs, as evidenced by recent excitement in the scientific community. Identifying bacterial promoters poses challenges, especially when genes are closely situated, potentially sharing regulatory elements. While the existence of these small

proteins could be demonstrated through proteomics, unraveling their function necessitates single-knockout experiments and a thorough phenotypic analysis.

In our study, most of the small ORFs evade detection by syntactic identification tools designed for bacteria. Situated within regions characterized by high accessory gene turnover, we can discount a pivotal role in the satellite's life cycle, proposing instead that they represent gene remnants. Notably, we have revised the characterization from 'highly questionable' to 'questionable' on line 118, aligning with the reviewer's suggestion. We concur with the reviewer that while we cannot completely rule out a potential role for these small ORFs in phage interference, further investigation is warranted."

Line 144: replace "has enabled" with "has allowed" to avoid repeating enabled.

Done

Lines 164-165: The authors suggest that their data is in line with rolling-circle replication. However, they also see an increase in circularisation over time (Figure 1C), which would suggest that there might be an element of replication of the circular form of the PICMI. Is this correct or an artifact of data presentation? See also comment about figure below.

In Figure 1C, the black bars illustrate the results of qPCR conducted with outward-directed primers. These primers amplify both the circular form of a single copy of PICMI and its concatemeric counterpart. This design makes it challenging to distinguish between the two forms by qPCR, and explain that we performed a southern blot. To provide clarity, we have added precision to the figure label, now denoting it as 'Circular and Concatemer.' The text is now: '*We then proceeded to investigate the kinetics of PICMI₁₁₅ activation following V115 infection by Φ 115pure. Inward-directed primers (as depicted in Fig. 1a) were utilized in qPCR to detect the empty integration site resulting from PICMI₁₁₅ excision. Outward-directed primers were employed to quantify the circularized PICMI₁₁₅, either as a single circular form or in concatemeric form.*'

Lines 177-178: Not sure that this is copy numbers. It rather sounds like the number of phage particles carrying PICMIs increased for the PICMI115-CmR compared to the wt PICMI.

We agree with the reviewer and changed the sentence by '*The introduction of the Cm^R cassette increased the percentage of PICMI₁₁₅ in phage particles (Supplementary Fig. 6).*'

The change is not insignificant (from 16% to 57% of the viral particle population) from Figure S6 and there now seems to be some level of interference with the phage that is being observed. For example, the phage titre now drops by almost 1-log unit compared to the titre observed when the wt PICMI strain was infected. PICMI titres from the marked PICMI seem to be similar to those obtained from the wt PICMI strain. This drop in phage titre disappears in either of the mutants that fail to induce the PICMI (indicating that it is related to PICMI induction) but is also absent on both USP1 and USP2 mutants. It seems unlikely that USP1 is related to this phenotype since it is upstream of the CmR suicide plasmid insertion. However, USP2 expression might be affected by the insertion of the plasmid, and this could lead to higher USP2 levels resulting in defence against the helper phage. Loss of USP2 restores phage titres. Note that in Figure 3C, USP2 is not induced as much as other genes by phage infection and the level of USP2 induction might well be what determines interference.

Thank you for your comprehensive analysis of our results. We have carefully considered these suggestions and present the figure below for further discussion. The reviewer suggests that over-expression of *up2*, caused by the insertion of the suicide CmR-vector at the end of this gene may lead to interference with phage Φ 115. This proposed interference is thought to explain the observed one-log decrease in the amount of phage genome and the increase in the PICMI-phage ratio from 15% to 57%. It is noteworthy that this interference is not observed in the Δ *up2* mutant.

Despite variations in the PICMI/phage ratio across experiments (as detailed below), the explanation faces a challenge from the Δ *up1* mutant. In this construct, where *up2* is not inactivated but could be over-expressed caused by the insertion of the suicide CmR-vector, we do not observe a decline in phage copy number, contrary to the expected interference scenario. To aid in the comprehension of this section of the materials and methods, we have included an additional figure 20 in supplementary. Furthermore, as the suicide CmR-vector is integrated downstream not upstream *up2* and *prim* genes, a polar effect seems unlikely.

Do you see this drop in copy numbers with the unmarked deletion mutants? Can you express different levels of USP2 prior to Phi115 infection to determine if this reduced the phage titre? Please note that I am not requesting this experiment for the manuscript to be accepted but rather as I found it an interesting observation.

While working with environmental strains in the laboratory, it's crucial to acknowledge the inherent variability in results between biological replicates, here concerning the PICMI/phage ratio. In the experiment detailed below, which addresses partially the reviewer's suggestions, we quantified Phage and PICMI copy numbers using DNA extracted from viruses produced from wild-type strains, both marked and unmarked, as well as unmarked mutants. PICMI was detected, in this case, using specific primers for each of the 6 genes rather than targeting the circular/concatemeric forms. While this approach allows for the confirmation of deletions, it does introduce more background noise due to the presence of contaminated host DNA (PICMI genes

from host genome). In this context, the PICMI/phage ratio averages at 4%, 6%, and 6% for wild-type (wt), wt-CmR, and $\Delta up2$, respectively, representing a subtle difference.

We appreciate the reviewer's suggestion of cloning the *up2* gene under the control of an inducible promoter to further evaluate its function as antiphage system including for the helper phage.

Lines 181-185: The formulation is a bit unclear and too dense to easily follow it. I would suggest rewording it, so the two infection experiments are separated and described individually as clearly different MOIs were required depending on the host strain.

Based on the comments from reviewer 2, we have revised this figure to depict both the phage titer (PFU) and the transductants titer (TFU). We acknowledge the importance of emphasizing the TFU/PFU ratio (or MOI, considering fixed bacterial CFU), as highlighted in our responses to reviewer 2. The sentence was changed to *‘Transductant Forming Units, TFU, were obtained using a lower titer of phages for $\Delta PICMI$ (10^5 Plaque Forming Unit, PFU, ml^{-1} ; Multiplicity of infection, MOI 0.01) compared to V511 (10^8 PFU ml^{-1} ; MOI 10). Consequently, the ratio between TFU and PFU was much higher with $\Delta PICMI$ ($\sim 6 \cdot 10^{-2}$) than with V511 ($\sim 10^{-5}$) (Fig. 2a)’*.

Lines 237-238: “This resulted from the increase of the number of copies of the PICMI115 genome and not necessarily through gene activation.” Please provide/refer to evidence.

We agree that we do not have evidence for this statement. We have changed the text to *‘An increase of transcripts for the remaining PICMI₁₁₅ genes (int, iolG and up2) was observed after 60 minutes. The latter might result from the increase in copy number of the PICMI₁₁₅ genome and not necessarily through the activation of gene expression.’*

Line 330: “Their minimalist gene” replace “Their” with “PICMI” to specify/clarify Done

Line 337: Define what is meant by “insignificant”. 50% can be quite significant biologically although it would not show up as statistically significant in phage titre data.

Changed to *'This observation aligns with earlier results for cf-PICI, in which production of the helper phages were also not significantly altered in the presence of a functional phage satellite¹¹'*

Line 447: “Φ115pure at MOI10 for 60 minutes” space missing between MOI and 10.

Done

Line 453: Greek symbols missing “the phage F115pure”.

We have edited all the missing symbols

Line 455-458: Greek symbols missing for phages and mutants.

Done

Line 459: “ILOG” should not be capitalised to be consistent with figure and gene nomenclature normally using non-capitalised spelling.

We changed IOLG by *iolg* everywhere for gene, and so we did for *up1* and *up2*.

Line 461: Replace “was detected by classical PCR and gel stained (upper panel) “ with “was detected by classical PCR and separation on agarose gel (upper panel)”

Done

Lines 463-477: Greek symbols missing for phages and mutants.

Done

Line 482: Replace “correspond” with “corresponds”.

Done

Line 517: Remove “upon request and”

Done

Line 520/561 (and throughout document): Capitalise “Vibrio” as a genus name.

We capitalize 'Vibrio' when referring to a specific species or strain. However, when discussing all species collectively, we use 'vibrio' (similar to the convention of 'bacteria' when referring to all bacteria).

Line 529: “Vibrio. Chagasii” either abbreviate or remove full stop. Check capitalisation.

Done

Line 572: correct formatting of “MgSO₄. 7H₂O

Done

Lines 589-591: Specify method for fold change calculation.

We now explain the method of calculation in the Materials and methods section and in the legend of Figures. *‘For all assays, three independent DNA samples (i.e., biological replicates) were tested under each condition. The number of copies for the phage, the empty integration site, and the circular/concatemeric PICMI115 was normalized by the number of copies of bacteria (gyrB) per sample. In the qRT-PCR analysis, the resulting copy numbers were further normalized to gyrA for each sample. Fold change was determined by comparing samples collected at specified time points to the input sampled immediately before adding the phage.’*

Lines 622-627: Please expand paragraph as the two approaches for generating mutants are not completely clear as described.

We trust that the enhanced explanation will meet the reviewer's expectations. Additionally, we have included a figure (Supplementary Figure 20) to aid in the comprehension of our methodologies. The text is now: *‘PICMI labelling (PICMI₁₁₅-Cm^R) was performed by cloning the 500bp end of up2 gene in the suicide plasmid pSW23T⁴². For the mutagenesis, two approaches were employed, depending on the target gene (Supplementary Fig. 20). Firstly, for gene deletion, 500bp fragments flanking the gene were cloned into the pSW7848T suicide plasmid⁴³. This vector encodes the ccdB toxin gene under the control of an arabinose-inducible and glucose-repressible promoter, P_{BAD}³⁸. Selection of the plasmid-borne drug marker on Cm and glucose resulted from integration of pSW7848T in the genome. The second recombination leading to pSW7848T elimination was selected on arabinose-containing media. Mutants were screened by PCR using external primers. For transduction experiments, mutants were marked by pSW23T insertion at the up2 end, as previously described. Secondly, for up2 and prim, a 500bp internal region of the gene was cloned into the suicide plasmid pSW23T (Supplementary Fig. 20). After conjugative transfer, plasmid-borne drug marker selection (Cm^R) resulted from the integration of pSW23T in the target region through a single crossing-over, leading to simultaneous gene inactivation and PICMI labeling. Integration of the suicide plasmid was confirmed by PCR using one primer in the plasmid and one in PICMI’.*

Line 623: change “and at the same time label the” to “and at the same time as labelling the”

Line 624: formatting of “ Δ prim and Δ UP2 -Cm^R”, are both mutants Cm^R-marked?
The whole paragraph has been revised (see above)

Line 668: change “tittering” to “titering”

Done

Line 672-674: A one hour incubation for a transduction seems very long considering that the phage replication cycle is less than that. Is there a particular reason for selecting this time frame and is it possible that transduction titres could be lower/higher because of repeated infection cycles during incubation?

The objective was to compare transduction efficiencies between the wild type and KO derivatives. Our chosen incubation time aligns with the estimated phage dynamics (Supplementary Fig. 7), indicating a peak particle production at 60 minutes. While literature methods, like cf-PICI, often

suggest a 30-minute incubation time, we acknowledge this discrepancy. We appreciate the reviewer's suggestion and plan to explore this kinetics aspect in our future work.

Line 753: *Vibrio cholerae* formatting in reference

We have now edited the references according to the Nature Communication format requirement

Figures:

Figure 1C: Unclear to what fold-change refers to. Is it to the values at 5 min, the average of the values at 5 min? Surely for excision and circularisation this should be 0 min to reflect baseline levels before infection. What is the phage baseline? I am struggling a bit with making sense of this data as it is presented. It might make more sense to present the excision data as percentage of the bacteria in which the PICMI has excised related to the total number of sites available would be the sum of those occupied (one of the grey + one of the black primers in figure 1A) or excised. Similarly, it is unclear whether the increase in circularisation is reflective of replication of circular forms of the PICMI or simply the result of different portions of the excised PICMI slowly circularising before entering rolling circle replication.

We apologize for the lack of clarity in the figure. We now provide comprehensive data, in terms of copy number, for each target (vibrio, phage, empty site, and circular or concatemeric PICMI) at each kinetic point, including t0 (Fig. 1c). We provide all data sources for this graph including the calibration range. Note the graph illustrates that the copy number of the empty site consistently remains below the copy number of the bacterial genome. We aim to delve into understanding this phenomenon in the future, exploring aspects such as excision occurring in only a fraction of cells or potential reversion of excision to integration. However, when comparing the number of copies of the circularized/concatemeric satellite to the empty site (multiplied by 10^3), and taking into account the results from both Illumina and Nanopore sequencing (where 15% of particles contain PICMI), it appears improbable that PICMI activation occurs in only a very small fraction of bacteria.

It's important to note that qPCR has its limitations, notably the emergence of a late Cq signal (26-30 cycles), introducing background noise and a potentially inaccurate estimate of copy presence. This background is now indicated by a hatched line. Additionally, as highlighted in the text, the black primers do not distinguish between the monomeric circular form and the concatemeric form. To address this, we conducted southern blots.

Lastly, for the sake of clarity and considering the allowance of eight items by Nature Communications, we have divided Figure 1 into two separate figures: Fig. 1 and 2.

Figure 2C. Contrast of agarose gels too high to see whether individual gels or a single gel. I don't think the fold change adds anything here and because of the no-change for *alpA*, it might be more confusing than informative. Figure S11 is far more informative here.

According to the recommendation of the two reviewers, we have change fig 2c and d (now Fig. 3c) by the figure S11. We have included the agarose gels in Fig. S11 with lower contrast. All uncropped gels are available in the data source. This information is retained to illustrate the complementation of *Δint* by the expression of *int* alone (as observed in Fig. 3c with *int/iolg*). We hope that this is OK

to give in Fig. 3a the Fold change (all qPCR results in data source), fully explain in the legend, to prevent overwhelming number of bar charts.

Figure 5B: I am missing the effect size in this graph. How strong is the interference for each pair? Also, I cannot see the control plasmid on the graph. Is this a relative ratio between the control and the UP2 plasmid within the same strain background? Does no phage production mean that the phage was unable to infect either the strain containing UP2 or gfp? I would suggest modifying this figure to incorporate effect sizes into to, potentially colour coded by fold-change, etc.

We have revised both the figure and its legend in response to the reviewer's feedback. Please note the following changes:

1. Figure 5b (now Figure 6b) provides a summary of the results obtained by testing positive interactions (i.e. phage killing) using the drop-deposition method for limiting dilutions of phages. In the figure, white squares represent resistant hosts, colored squares represent phage-sensitive hosts, and black squares indicate no modification in phage production by UP2, while red squares signify a reduction in production. This approach allows for testing a large number of combinations and identifying sensitive phages.
2. The set of combinations influenced by UP2 (versus the GFP control) is depicted in Figure S18.
3. Further details of the UP2 effect were measured by infection in liquid culture to assess the titer of the progeny, and the result is presented in Fig. 6c. Notably, two phage-non-original host combinations were excluded from this analysis, as their infection was already considerably weaker than the others, preventing the execution of a liquid experiment under comparable conditions. The ratio of phage titer in UP2 to GFP strains is provided above the bar chart.

Figure S6: Can you explain why you can still detect PICMI-115 particles in the Δ PICMI115 mutant lysate? I would expect them to be not-detected in a clean lysate. Is there carryover, contamination or a limitation of the assay that does not allow to detect below a certain level of copies? What is the limit of detection of the assay? Can this be indicated in the figure? Please also show the reference curves used for calibration of the copy numbers.

This is clearly a limit of qPCR assay now indicated by a hatched line, as Southern blot in 3b and nanopore sequencing (Table S1) indicate that there is no PICMI in particles produced by Δ PICMI₁₁₅.

Figure S9: Define the bar to the left of the figure. Increasing dilutions/decreasing concentrations?

We added 'free phage dilution' in the figure.

Reviewer #2

Congratulations to the authors; this is a fascinating piece of work in which the authors have characterized a new family of satellites in Vibrionaceae, named the phage-inducible chromosomal minimalist Island (PICMI). The text is pleasant to read, I enjoyed following the rationale and logic of their experiments after their observations on the two contigs that appeared on their initial exploration on *V. chagasii* phages.

The manuscript provides a convincing argument as to why this family should be considered related to the PICMI umbrella yet kept apart with a different name due to their unique mutualistic lifestyle. The authors neatly show that PICMIs are capable of being induced and transferred by lytic phages, while altering very little the production of their inducing phage. Importantly, their distribution amongst Vibrionaceae species suggests they could greatly impact phage diversity and play an important role in horizontal gene transfer as other satellites do.

We express our sincere gratitude to the reviewer for their encouraging and constructive feedback. We concur with its assessment that this initial exploration of PICMI presents numerous exciting prospects, which we eagerly anticipate exploring in future research endeavors. We are pleased to become a part of the satellite community.

I would like to suggest a change in the title: “Phage inducible chromosomal minimalist island (PICMI), a new family of small marine satellites of virulent phages.”

We changed the title according to the reviewer recommendations (now 15 words).

I have only minor comments and suggestions that the authors should address:

General comment:

Figures with bar charts. Please use standard deviation and appropriate statistical test to validate significance. This is a standard form for the journal.

We now describe in each figure legend that we use SEM, appropriate for our data set. We also use statistical tests when significance is tested (in Figure legends).

When reporting phage titer, I personally find it easier to read when the figure depicts PFU/ml and not fold change, as I get an overview from the initial phage titer and can be related with the transduction titer (TFU/ml). I would leave this to the editor and consider if this could help the reader.

We now give the titer of phage at t0 and t60 minutes post infection in Figure 2c and control without bacteria vs titer of phage at 60 min post infection in Fig. 6 a and c. We however discussed below the necessity to give the ratio of TFU on the phage titer in Figure 2a as transductants have been observed in V115 with 1000 less phages than V511.

PICMI-like elements distribution: These elements clearly tend to be minimal. Were there similar satellites found in other bacterial species?

We did not find intact PICMI-like elements in other bacteria (state line 269).

Identification of defense system: I really like the authors approach to identify and test such genetic modules contained within the PICMI. However, in some instances, the fold change reduction on phage titer is just a 100-fold and this works with few phages as seen in Figure 5. Would it be possible that UP2 mitigates infection in a different manner by controlling a specific set of genes in the host that cannot be use by such phages? I think the authors should provide more information as to why UP2 should be consider an anti-viral module and why this would exclusively protect from non-helper phages.

We now showcase the phage titer obtained using GFP or UP2 strains and the UP2/GFP ratio. Additionally, we have incorporated several discussion sentences into the text, carefully avoiding undue speculation based on the existing data. *‘Within this phage family, UP2 had no impact on the helper phage Φ 115, whereas the titers of the remaining phages were reduced, ranging from partial (i.e. Φ 120) to nearly complete abrogation (i.e. Φ 511) (Fig. 6c). The observed variation in UP2-mitigated infection suggests that coevolution may have already influenced this mechanism. This influence could manifest either through conferring an escape mechanism on the phage side or by triggering additional defense systems on the host side’.*

And we conclude this section by *‘We conclude that PICMI protects the bacterial host from non-helper phage. This protection relies at least in part on a novel UP2 defense system, whose activity is dependent on the host and phage genetic background. This highlights that certain defense systems exhibit such specificity that they can only be effectively studied within a limited genetic context, encompassing both the phage and its host. This underscores the importance of utilizing collections derived from natural populations and cross infection matrices to enhance the relevance and applicability of findings.’* Highlighting the benefit of our approach in identifying and testing new defense systems.

Discussion:

I would encourage the authors to make a comment regarding the packaging of the PICMI and how this new family can be a significant aspect to pathogenicity.

We discuss these points in the results section as well as in the discussion. Having no gene candidate for any of these functions, it is still a limited point to discuss.

Could UP1 have a role in packaging?

Given that the deletion of UP1 has no discernible impact on the presence of PICMI in particles and transduction efficiency, this appears to be unlikely, at least under laboratory conditions.

PICIs in pathogenic strains tend to carry toxins, antibiotic resistance and/or phage interference genes. Is there any evidence that PICMIs contribute to the virulence of their host? Are PICMIs associated with strains that can cause disease?

We have a limited amount of indirect evidence to address this question. Firstly, all tested *V. chagasii* in oyster experimental infections have demonstrated virulence, even in the absence of PICMI. Conversely, a non-virulent strain of *V. aestuarianus* carries PICMI.

Additionally, we conducted an analysis of accessory gene annotations on PICMI of Type A, B or C (172 in total), by extracting the proteomic regions that were inferred to correspond to their genomes (i.e., all the ORFs between the integrase and *fts*). AMRFinder detected no antibiotic resistance genes in PICMI. DefenseFinder detected 13 anti-phage defense systems in 12 PICMIs (all of them of Type C). No individual genes from other defense systems were detected. An additional search using the virulence factor database (VFDB) detected genes homologous (albeit with <35% identity) to known virulent factors in 9 additional PICMI (8 Type C and 1 Type B). These results are shown in Table S2 and 3 and below:

Genome	PICMI Type	AMRFinder	DefenseFinder	VirulenceFactors
GCA_013729995.1_ASM1372999v1	Type C	--	AbiJ	--
GCA_013730455.1_ASM1373045v1	Type C	--	AbiJ	--
GCA_009874545.1_ASM987454v1	Type C	--	--	VFG044083
GCA_013738635.1_ASM1373863v1	Type C	--	--	VFG000077
GCA_012967125.1_ASM1296712v1	Type C	--	--	VFG000077
GCA_002877655.1_ASM287765v1	Type C	--	Retron_II	
GCA_001718055.1_ASM171805v1	Type C	--	--	VFG000077
GCA_000147055.1_ASM14705v1	Type C	--	--	VFG000077
GCA_009372095.1_ASM937209v1	Type C	--	--	VFG048229
GCA_002877655.1_ASM287765v1	Type C	--	Retron_II	--
GCA_015818055.2_PDT000168087.2	Type C	--	RM_Type_I	--
GCF_009372095.1_ASM937209v1	Type C	--	--	VFG048229
GCA_024043805.1_PDT001333937.1	Type C	--	RM_Type_II	--
GCF_006374125.1_ASM637412v1	Type C	--	RM_Type_I	--
GCF_006374125.1_ASM637412v1	Type C	--	Septu	--
GCF_024745165.1_ASM2474516v1	Type C	--	RM_Type_II	--
GCF_004104355.1_ASM410435v1	Type C	--	PARIS_I	--
GCA_019853935.1_ASM1985393v1	Type B	--	--	VFG048229
GCA_014918795.1_ASM1491879v1	Type C	--	RM_Type_I	--
GCA_024034875.1_PDT001334863.1	Type C	--	RM_Type_I	--
GCF_009874545.1_ASM987454v1	Type C	--	--	VFG044083
GCA_023814385.1_PDT001331953.1	Type C	--	RM_Type_I	--
GCA_023817365.1_PDT001333935.1	Type C	--	RM_Type_II	--

Can the newly characterized defense system UP2, in PICMI115 influence other MGEs, such as plasmids?

As we successfully complemented the mutant by expressing genes from a plasmid (P15A oriV), the element does not appear to be involved in defense against plasmids. However, a precise answer to this question would require testing various plasmids. Considering the extreme specificity of UP2 for the P115 family, UP2-mediated plasmid defense seems quite unlikely.

Comments:

Line 25 and whenever referred to phage-inducible chromosomal islands, keep the hyphen for consistency with other related works.

Done

Line 50-52. I would like the authors to revise such paragraph as it lacks some information. PICIs are more abundant and spread in the bacteria phyla than other satellites, including species in Firmicutes, Enterobacteriaceae, Bacillaceae, and Gammaprotobacteria. Cf-PICIs are a bit less abundant and mainly in Enterobacteriaceae, Lactobacilli, Enterococci, Pasteurella, Bacillaceae, Streptococci and Morganellaceae. P4s are mainly in Enterobacteriaceae, Yersiniaceae, Erwiniaceae, Hafniaceae and Pectobacteria.

We added these informations in the paragraph, now: *‘Phage-inducible chromosomal islands (PICIs)^{3,10} are the most widespread satellites, being found in many Firmicutes and Gammaprotobacteria. They are closely related with the capsid-forming PICIs (Cf-PICIs)^{11,12}, which are less abundant and primarily found in Enterobacteriaceae, Lactobacilli, Enterococci, Pasteurella, Bacillaceae, Streptococci, and Morganellaceae. P4-like satellites are prevalent in Enterobacteriaceae, Yersiniaceae, Erwiniaceae, Hafniaceae, and Pectobacteria¹³. In contrast, phage-inducible chromosomal islands-like elements (PLEs) are identified exclusively in Vibrio cholerae^{14,15}’.*

Line 56. The authors explained some common features about satellite induction, but what about satellites induced by other satellites. Check Haag et al. (2021) Nat Microbiology.

In this introduction, adhering to Nature Communication’s concise standards, we present essential knowledge for our study. The induction of satellites by others appears tangential to our narrative.

Line 75. I share the views of the authors regarding the complex relation between bacteria, the phage, and the satellite. What about symbiosis? Could we start considering such elements as symbionts who play important role in evolution and adaptation?

We interpret the term ‘symbiosis’ used by the reviewer to denote mutualism, a point previously addressed in the preceding sentence (i.e. “some P4-like satellites²⁰ and PICI²¹ encode hotspots of antiviral systems protecting both the bacterial host and their helper phages from competing phages and other mobile genetic elements. The associations of known phage satellites thus range from pure parasitism to mutualism in relation to their bacterial and phage hosts.)

Line 84. Change “we addressed this challenge...” to “we addressed these challenges....”

Done

Line 109. Change “a putative regulator” to “a putative transcriptional regulator”. Then this will make more sense when reaching the AlpA experiments later.

We prefer to maintain the term ‘putative regulator’ for AlpA. Previous studies attribute dual functions to AlpA—transcriptional regulation and/or excisionase. Our findings, however, lack

evidence supporting the idea that AlpA from PICMI is a transcriptional regulator; instead, it appears to act on the excision of the element.

Line 114. Is there a reference for Phanotate?

We added the reference 'PHANOTATE: a novel approach to gene identification in phage genomes' from McNair 2019.

In lines 144-150. This paragraph is exciting. I wonder if the authors could expand on a few aspects related to packaging. Typically, we found that the type of PICIs or satellites have similar packaging mechanism as their related helper phage (this being cos or pac). Is the inducing phage a cos or pac type? If the inducing phages have a TerS, can such protein be employed by the PICMI?

We already gave this information in the first paragraph: Our cultivation-enabled model system has allowed us to dissect the various steps in the PICMI's life cycle: (i) excision, (ii) replication, (iii) packaging (iv) transduction to a new host. Of these, we were not able to identify *cos* or *pac* packaging sites in the helper phage genome, or any homologs of genes involved in redirecting packaging that are characteristic of other satellite families (i.e., *terS*, *sid*, *ppi*). Agreeing with the reviewer, we acknowledge that a prospective aspect of this study is to comprehend the packaging of PICMI, aiming to unveil shared or distinctive features among various satellite families

Line 171. I guess the authors could also verify and mentioned a few defense mechanisms using Padloc or DefenseFinder. Regarding *fis*, this is more curiosity, but what significance has *fis* and does it play a major function in the *Vibrio* species?

The genome of the V511 strain, akin to other vibrios in our collection, harbors established anti-phage defense systems, along with likely unidentified ones. Using defense finder version version 1.0.8 we found that a Dnd and a Lamassu systems present in V511 and absent in V115. Now we also found 160 genes that are specific to V511, for the vast majority encoding unknown function and localized in diverse genomic islands. Without a genetic validation of these systems' functions, enumerating potential defensive elements appears speculative and unrelated to the findings presented in this article. To satisfy the reviewer we add line 180: '*Specifically, the Dnd and Lamassu defense systems were identified in V511 using Defense-Finder³¹, while they were notably absent from V115.*'

In relation to *fis*, we provide detailed information and a reference for this gene: '*Surrounded by two 17 bp direct repeats, this element integrates downstream of the fis gene (Fig. 1a). The fis gene encodes a DNA-binding protein that plays a crucial role in the efficient excision of phage lambda from the Escherichia coli chromosome²⁷.*' It is tempting to speculate that *fis* provide a way for the PICMI to manipulate excision (or lack of excision) of the phage, an hypothesis that we will test in the future.

Figure 1. *fis* is mentioned often in the text, but I have no background on its context. It would be a nice addition a small description of what *fis* and *zntR* are for its host, and if there is a relation with phage integration. Are there any phages integrated there?

We now provide details about *fis* (see above). *ZntR* and *fis* are now defined in the Fig. 1 legend. *ZntR* is adjacent to *fis* in *V. chagasii* and does not belong to the PICMI element. Moreover, when

we looked for phage structural genes in the vicinity of these sites (25Kbs in both directions) using PHANNNS, we found no homologous for such genes.

Line 181-187 and Figure 1E. I appreciate the authors experimental design by normalizing the number of transductants by the number of phages, however this can be simplified by indicating the number of TFU/ml that were obtained.

We encountered an issue with this request. The transduction tests were conducted at serial MOIs for both Δ PICMI and V511. It was observed that transductants were countable at a lower MOI for Δ PICMI compared to V511 (refer to the data sources, now included in the data source folder).

	Experiment 1			Experiment 2			Experiment 3		
	PFU/ml	TFU/ml	TFU/PFU	PFU/ml	TFU/ml	TFU/PFU	PFU/ml	TFU/ml	TFU/PFU
Delta PICMI	1,00E+06	7,40E+03	7,40E-03	1,00E+06	4,00E+03	4,00E-03	1,00E+06	7,00E+03	7,00E-03
V511	1,00E+09	1,20E+03	1,20E-06	1,00E+09	9,00E+02	9,00E-07	1,00E+09	1,30E+03	1,30E-06

Therefore, if we only present the TFU/ml (in grey) this could lead to the impression of similar transduction efficiency for V511. This might be misleading, as a higher phage titer in the input was necessary to achieve these transductants (10^+9 versus 10^+6). So as an alternative will be to show both the PFU and TFU/ml and highlight the average TFU/PFU.

Line 195-196 and Figure 1F. I would rather the authors to employ PFU/ml and remember to highlight what was the recipient or propagating strain.

We revised Figure 1f using PFU/ml and discriminated the titer of phage Φ 115 pure (always produced from V115 delta PICMI), added to the bacterial culture at t0 and 60 minutes post-infection of each of the four strains. Additionally, we applied a black/grey scale to color each dot to represent the respective biological replicates. The recipients are indicated below the histogram.

Line 210. The leaky expression of alpA by the PBAD promoter seems to be the caused to this early excision, suggesting that the mechanism could be dose dependent.

We acknowledge this observation and plan to formally test it in the future

Figure 2C. The southern blot seems to indicate that prim deletion could have an impact on phage 115. Any comments as to why the band look so different from the rest? I would expect to have similar intensities for bands with int, OLG, alpA, prim and PICMI115 deleted if truly the PICMI does not impact phage reproduction.

This DNA originates from viral particles produced by the different mutants, not from bacteria. The reduced intensity of the band in the Southern blot results from a lower quantity of DNA loaded on the gel (refer to the staining gel, also provided in the data source as an uncropped gel image). This discrepancy may arise from the normalization of all DNAs based on nanodrop quantification, which is known to be imprecise for genomic DNA.

I really like Figure S11. Maybe this one can substitute the lower section of Figure 2C, or be another panel as Figure 2D.

To satisfy both the reviewers 1 and 2 Figure 2C and D have been replaced by Figure S11.

Please check the math symbols for this paragraph (Figure legend 2) as some have may not be transferred adequately.

We have check and edit all symbols that was lacking in Figure legends (also mentioned by reviewer 1)

Regarding AlpA; the authors compared its structure with Xis and TorL. However, I would like them to discuss the difference of this AlpA to the AlpA expressed by PICIs and other MGEs such as ICEs. Is this an annotation problem that should be addressed by the community employing better prediction software?

We found some homology towards some alpA in some other satellites, and we are currently investigating that.

Lines 224-241. It seems as the seven phages are related to phage 115. Is phage 27 much different from the others? This could suggest what and how the PICMI has been induced. Possibly the differences between 27 and 115 could point towards the mechanism from which alpA is activated.

We acknowledge the excitement of this perspective. However, it is important to note that after identifying SNPs in P27, we will need to genetically demonstrate that these modifications in the phage alter the activation of PICMI, and explore the mechanism of a putative activator encoded by the helper phage. This constitutes an entire PhD project, and we are currently in the process of recruiting a student to undertake this research.

Lines 282-286. Any comments on what UP1 function could be?

The UP1 gene encodes an unknown function, and despite its distribution in various PICMIs, experimental testing indicates it is not essential for their activity under current conditions.

Therefore, assigning a specific role for this protein remains challenging with our current knowledge.

Lines 313-318. Could there be other modules at work to hinder the interference? Maybe other elements within the host genome

We concur that phage defense systems have been demonstrated as additive and synergistic. We agree it is also plausible to speculate that some systems may be antagonistic, hindering interference.

Line 326. “These results...”

Done

Line 350 “...this finding suggest a potential trade-off..”

Done

Line 498 “...GFP strains are...”

Done

Table S3 is missing from manuscript.

Thank you , we added the reference for this tableS3 in Figure 5 legend

Reviewer #1 (Remarks to the Author):

The authors have addressed all my previous remarks. I am happy with the article proceeding to publication.

I have only one minor comment related to a typo in the figures:

Figure 1c Y-axis: Replace "Copies number" with "Copy number"

Reviewer #2 (Remarks to the Author):

Congratulations to the authors; they have addressed all my comments and suggestions. I enjoyed reading again the manuscript and I can see it has improved both in presentation and narrative.

Minor details:

Line 360. Please check this sentence as it is a bit confusing.

Did you mean?

"The PICMI family is among the smallest of phage satellites, with PICMI115 being the smallest satellite element with characterized/described activity."

Line 437. Please change Luria-Bertani to Lysogeny Broth.

Figure 1. I find confusing to reference other Figures within another figure. Try to omit this.

Figure 2. Check colours on bar and labels (I think there was a modification intended)

Aside from these, I have no further comments.

REVIEWERS' COMMENTS

Reviewer #1 (Remarks to the Author):

The authors have addressed all my previous remarks. I am happy with the article proceeding to publication. I have only one minor comment related to a typo in the figures:

Figure 1c Y-axis: Replace “Copies number” with “Copy number”

Done, see below

c

Reviewer #2 (Remarks to the Author):

Congratulations to the authors; they have addressed all my comments and suggestions. I enjoyed reading again the manuscript and I can see it has improved both in presentation and narrative.

Minor details:

Line 360. Please check this sentence as it is a bit confusing. Did you mean? “The PICMI family is among the smallest of phage satellites, with PICMI₁₁₅ being the smallest satellite element with characterized/described activity.”

Changed by “The PICMI family is among the smallest of phage satellites, with PICMI₁₁₅ being the smallest satellite element with characterized activity.”

Line 437. Please change Luria-Bertani to Lysogeny Broth.

Done

Figure 1. I find confusing to reference other Figures within another figure. Try to omit this.

We suppressed reference to other figures

Figure 2. Check colours on bar and labels (I think there was a modification intended)

We hope the color code is clear now, see below.